# PROBLEM-PARAMETER-FREE FEDERATED LEARNING

**Wenjing Yan**[1], **Kai Zhang**[2], **Xiaolu Wang**[3], **Xuanyu Cao**[4*]
[1]The Chinese University of Hong Kong    [2]The Hong Kong University of Science and Technology
[3]East China Normal University    [4]Washington State University
{wjyan}@ie.cuhk.edu.hk, {kzhangbn}@connect.ust.hk,
{xiaoluwang}@sei.ecnu.edu.cn, {xuanyu.cao}@wsu.edu

## ABSTRACT

Federated learning (FL) has garnered significant attention from academia and industry in recent years due to its advantages in data privacy, scalability, and communication efficiency. However, current FL algorithms face a critical limitation: their performance heavily depends on meticulously tuned hyperparameters, particularly the learning rates. This manual tuning process is challenging in federated settings due to data heterogeneity and limited accessibility of local datasets. Consequently, the reliance on problem-specific parameters hinders the widespread adoption of FL and potentially compromises its performance in dynamic or diverse environments. To address this issue, we introduce PAdaMFed, a novel algorithm for nonconvex FL that carefully combines adaptive stepsize and momentum techniques. PAdaMFed offers two key advantages: 1) it operates autonomously without relying on problem-specific parameters, and 2) it manages data heterogeneity and partial participation without requiring heterogeneity bounds. Despite these benefits, PAdaMFed provides several strong theoretical guarantees: 1) it achieves state-of-the-art convergence rates with a sample complexity of $\mathcal{O}(\epsilon^{-4})$ and communication complexity of $\mathcal{O}(\epsilon^{-3})$ to obtain an accuracy of $\|\nabla f(\boldsymbol{\theta})\| \leq \epsilon$, even using constant learning rates; 2) these complexities can be improved to the best-known $\mathcal{O}(\epsilon^{-3})$ for sampling and $\mathcal{O}(\epsilon^{-2})$ for communication when incorporating variance reduction; 3) it exhibits linear speedup with respect to the number of local update steps and participating clients at each global round. These attributes make PAdaMFed highly scalable and adaptable for various real-world FL applications. Extensive empirical evidence validates the efficacy of our approach.

## 1 INTRODUCTION

Federated learning (FL) has emerged as a promising paradigm for machine learning, allowing multiple clients to collaboratively train a model without sharing raw data. Since its introduction by McMahan et al. (2017), FL has garnered substantial attention from both academia and industry. Major conferences such as NeurIPS, ICML, and ICLR have witnessed a proliferation of FL-related research, addressing critical challenges including communication efficiency (Chen et al., 2021; Sattler et al., 2019), privacy preservation (Wei et al., 2020; Mothukuri et al., 2021), heterogeneity management Li et al. (2020); Karimireddy et al. (2020b); Wang et al. (2020), and partial and asynchronous participation (Wang et al., 2024; Xu et al., 2023).

Despite significant advancements, current FL algorithms face a critical limitation: their performance heavily depends on meticulously tuned hyperparameters, particularly the learning rates. This tuning process typically requires extensive computational resources and problem-specific knowledge, such as smoothness parameters, heterogeneity bounds, stochastic gradient variances, and initial optimality gaps. For instance, MIME (Karimireddy et al., 2020a) relies on smoothness constants and data heterogeneity bounds for stepsize determination, FedProx (Li et al., 2020) requires careful adjustment of a proximal term based on data heterogeneity, and FedDyn (Acar et al., 2021) demands tuning of a regularization parameter contingent on problem characteristics. Furthermore, algorithms such as FedADT (Gao et al., 2023) and FedAMS (Chen et al., 2020) necessitate problem-specific

---

*Corresponding Author.

parameters to establish minimum communication rounds, which must exceed thresholds associated with smoothness constants and other problem characteristics.

This reliance on problem-specific parameters poses several critical challenges. First, it impedes the widespread adoption of FL by complicating deployment and requiring expertise for hyperparameter tuning (Mostafa, 2019; Deng et al., 2020). Second, it potentially compromises the performance of FL in dynamic or diverse environments where data distributions may evolve (Reddi et al., 2020; Koloskova et al., 2020). Last, accurately estimating these parameters is often infeasible due to the distributed nature of data and the inherent privacy constraints of FL (Konečnỳ et al., 2016).

Recent research has attempted to address this issue through adaptive stepsize methods. FedOpt (Reddi et al., 2020) incorporates adaptive optimization techniques like AdaGrad, Adam, and Yogi into FL, demonstrating improved convergence compared to FedAvg (McMahan et al., 2017). Fed-Nova (Wang et al., 2020) introduces a normalization technique that effectively mitigates objective inconsistency caused by partial client participation and data heterogeneity. FedBN (Li et al., 2021) employed local batch normalization to alleviate feature shift before averaging models, outperforming both classical FedAvg (McMahan et al., 2017) and FedProx (Li et al., 2020) for non-IID data. While these approaches show promise, they still require careful tuning of global learning rates.

Momentum is a technique that mitigates data heterogeneity and accelerates gradient descent by maintaining a velocity vector of gradient history. Recent studies have investigated the combination of adaptive methods with momentum to leverage both advantages. Hsu et al. (2019) proposed Fe-dAvgM, which incorporates a server-side momentum term into the FedAvg algorithm, demonstrating enhanced convergence rates and robustness against data heterogeneity. Wang et al. (2019) developed SlowMo, a momentum-based method that employs two nested loops to facilitate faster convergence in distributed optimization. FedAMS (Chen et al., 2020) utilizes adaptive moment methods on both the server and client sides to address data heterogeneity. MIME (Karimireddy et al., 2020a) combines client and server momentum to enhance convergence. Wu et al. (2023) introduced FAFED, a momentum-based variance reduction scheme integrated with an adaptive matrix, achieving the best-known sample and communication complexity when utilizing diminishing stepsizes. Nevertheless, these methods still necessitate fine-tuning multiple hyperparameters, limiting their practical applicability.

A very recent contribution, FedSPS (Sohom Mukherjee, 2024), claims to be the first fully locally adaptive method for FL with minimal hyperparameter tuning. While promising, this approach relies on stringent assumptions of bounded gradients and bounded data heterogeneity. Moreover, it fails to converge to optima with constant stepsizes and requires adjustment of a maximum stepsize threshold based on the smoothness parameter, maintaining a degree of hyperparameter dependence. Therefore, there is a critical need for more robust and adaptive FL algorithms capable of operating effectively across diverse scenarios without relying on problem-specific parameters.

## 1.1 MAIN CONTRIBUTIONS

This paper addresses these critical limitations of FL by proposing a novel approach that eliminates the need for problem-specific hyperparameter tuning while effectively handling arbitrary heterogeneous data. Our method is based on a careful combination of adaptive stepsize and momentum techniques. The adaptive stepsize mechanism dynamically adjusts local learning rates against client update heterogeneity, while the momentum component provides stability under partial participation and accelerates convergence in the nonconvex landscape. To the best of our knowledge, this is the first algorithm to achieve such parameter-agnostic adaptation in nonconvex FL with a state-of-the-art convergence guarantee. Our main contributions are summarized as follows.

1) We introduce PAdaMFed, a problem-specific **P**arameter **A**gnostic algorithm for nonconvex FL based on **ada**ptive stepsizes and client-side **M**omentum. PAdaMFed offers several significant advantages:

   - *Independent of problem-specific parameters*: PAdaMFed operates autonomously without relying on any problem-specific parameters such as smoothness constants or stochastic gradient variance. All stepsizes in our approach are explicitly determined by system-defined constants, including the number of participating clients, local updates, and communication rounds.

- *Robustness to arbitrary heterogeneous data*: PAdaMFed inherently manages data heterogeneity without requiring any heterogeneity bounds among clients while accommodating partial client participation. This feature enhances its scalability and adaptability in real-world scenarios where client data can be highly diverse and unpredictable, and full participation may not always be feasible due to resource constraints or device availability.

2) We provide a rigorous theoretical analysis of PAdaMFed, demonstrating its state-of-the-art performance:

- PAdaMFed achieves a sample complexity of $\mathcal{O}(\epsilon^{-4})$ and communication complexity of $\mathcal{O}(\epsilon^{-3})$ to obtain a $\|\nabla f(\boldsymbol{\theta})\| \leq \epsilon$ accuracy for nonconvex FL problems, even using constant learning rates.
- The complexities are further improved to the best-known sample complexity of $\mathcal{O}(\epsilon^{-3})$ and communication complexity of $\mathcal{O}(\epsilon^{-2})$ when incorporating variance reduction.
- PAdaMFed exhibits linear speedup with respect to the numbers of local update steps and participating clients in each global round.

Notably, these theoretical results are obtained under minimal assumptions, requiring only $L$-smoothness of loss functions and unbiased stochastic gradients with bounded within-client variance. This represents a significant advancement over existing FL algorithms, which typically necessitate constraints such as data heterogeneity bounds (Li et al., 2021; Wu et al., 2023), diminishing stepsizes (Wu et al., 2023; Sohom Mukherjee, 2024), or fail to achieve the best-known convergence rates (Liang et al., 2019; Alghunaim, 2024).

3) We conduct empirical evaluations to validate our theoretical findings and the efficacy of our algorithms. Our methods are compared against several established baselines, including FedAvg (McMahan et al., 2017), SCAFFOLD (Karimireddy et al., 2020b), and SCAFFOLD-M (Cheng et al., 2024) . Extensive numerical evidence demonstrates the superiority of our approaches in not only runtime efficiency but also stepsize robustness.

## 2 PROBLEM SETUP

We consider an FL system where $N$ clients collaboratively train a common learning model $\boldsymbol{\theta} \in \mathbb{R}^d$ under the coordination of a parameter server. Let $\boldsymbol{\xi}_i$ represent a random sample of client $i$ drawn from its local data distribution $\mathcal{D}_i$. The loss function associated with client $i$ is given by $f_i(\boldsymbol{\theta}) := \mathbb{E}_{\boldsymbol{\xi}_i \sim \mathcal{D}_i}[F(\boldsymbol{\theta}; \boldsymbol{\xi}_i)]$, where $F(\boldsymbol{\theta}; \boldsymbol{\xi}_i)$ is the stochastic loss of client $i$ over sample $\boldsymbol{\xi}_i$. The objective of the FL system is to minimize the global loss function across all clients, defined as:

$$\min_{\boldsymbol{\theta} \in \mathbb{R}^d} f(\boldsymbol{\theta}) := \frac{1}{N} \sum_{i=1}^{N} f_i(\boldsymbol{\theta}) \quad \text{where} \quad f_i(\boldsymbol{\theta}) := \mathbb{E}_{\boldsymbol{\xi}_i \sim \mathcal{D}_i}[F(\boldsymbol{\theta}; \boldsymbol{\xi}_i)].$$

In a federated setting, clients collaboratively train a global model, but the raw data of each client is never shared with the server and other clients.

Denote by $\|\cdot\|$ the $\ell_2$ norm. We make the following standard assumptions.

**Assumption 1** (Sample-Wise Smoothness). *Each sample-wise loss function $F(\boldsymbol{\theta}; \boldsymbol{\xi}_i)$ is $L$-smooth, i.e., $\|\nabla F(\boldsymbol{\theta}; \boldsymbol{\xi}_i) - \nabla F(\boldsymbol{\delta}; \boldsymbol{\xi}_i)\| \leq L\|\boldsymbol{\theta} - \boldsymbol{\delta}\|$ for all $\boldsymbol{\theta}, \boldsymbol{\delta} \in \mathbb{R}^d$, $i \in \{1, \ldots, n\}$, and $\boldsymbol{\xi}_i \sim \mathcal{D}_i$.*

The Sample-Wise Smoothness Assumption 1 implies the following standard smoothness condition.

**Assumption 2** (Standard Smoothness). *There exists $L > 0$, such that each loss function $f_i$ is $L$-smooth, i.e., $\|\nabla f_i(\boldsymbol{\theta}) - \nabla f_i(\boldsymbol{\delta})\| \leq L\|\boldsymbol{\theta} - \boldsymbol{\delta}\|$ for all $\boldsymbol{\theta}, \boldsymbol{\delta} \in \mathbb{R}^d$ and $i \in \{1, \ldots, N\}$.*

We emphasize that our original PAdaMFed algorithm is based on the Standard Smoothness Assumption 2. The slightly more stringent Assumption 1 is required when using variance reduction to further accelerate convergence.

**Assumption 3** (Stochastic Gradient). *There exists $\sigma \geq 0$ such that for any $\boldsymbol{\theta} \in \mathbb{R}^d$ and $i \in \{1, \ldots, N\}$, $\mathbb{E}_{\boldsymbol{\xi}_i}[\nabla F(\boldsymbol{\theta}; \boldsymbol{\xi}_i)] = \nabla f_i(\boldsymbol{\theta})$ and $\mathbb{E}_{\boldsymbol{\xi}_i}\|\nabla F(\boldsymbol{\theta}; \boldsymbol{\xi}_i) - \nabla f_i(\boldsymbol{\theta})\|^2 \leq \sigma^2$, where $\boldsymbol{\xi}_i \sim \mathcal{D}_i$.*

Assumption 3 ensures that the stochastic gradient $\nabla F(\boldsymbol{\theta}; \boldsymbol{\xi}_i)$ is unbiased and has bounded within-client variance, which is standard in stochastic optimization.

We consider nonconvex FL problems with data heterogeneity among clients, where the local data distributions $\mathcal{D}_i \neq \mathcal{D}_j$ for any $i \neq j$. When addressing heterogeneous data, most existing approaches, such as SCAFFOLD (Karimireddy et al., 2020b), FedProx (Li et al., 2020), FedAMS

(Chen et al., 2020), and MIME (Karimireddy et al., 2020a), require an upper bound on gradient dissimilarity, i.e., there exist constants $B, \sigma_h^2 > 0$ such that

$$\frac{1}{N} \sum_{i=1}^{N} \|\nabla f_i(\boldsymbol{\theta})\|^2 \leq B \|\nabla f(\boldsymbol{\theta})\|^2 + \sigma_h^2 \text{ for all } \boldsymbol{\theta} \in \mathbb{R}^d. \tag{1}$$

This assumption simplifies the mathematical analysis of those FL approaches and ensures their algorithmic performance. However, it may not hold in scenarios where data across clients exhibit significant and unpredictable variations, thus compromising the robustness of FL.

Additionally, existing FL algorithms typically rely on problem-specific parameters to determine their stepsizes, including the smoothness constant $L$, gradient variance $\sigma^2$, and heterogeneity bounds $B$ and $\sigma_i^2$. The smoothness constant, which characterizes the Lipschitz continuity of gradients, is generally a global property requiring knowledge of the entire dataset. Similarly, quantifying data heterogeneity across clients necessitates a comprehensive understanding of the differences between local data distributions. However, in FL settings where raw data sharing is prohibited and only model updates are exchanged, obtaining precise measurements of these parameters is computationally prohibitive and may compromise FL's privacy guarantees.

In the subsequent section, we present an algorithm that is independent of problem-specific parameters and capable of handling arbitrarily heterogeneous data, thereby eliminating the requirement of the heterogeneity bound (1).

## 3 ALGORITHM DEVELOPMENT

---

**Algorithm 1** PAdaMFed: A Problem-Parameter-Agnostic Algorithm for Nonconvex FL

---

1: **Require:** initial model $\boldsymbol{\theta}^0$, control variates $\boldsymbol{c}_i^{-1} = \frac{1}{K} \sum_{k=0}^{K-1} \nabla F\left(\boldsymbol{\theta}^0; \boldsymbol{\xi}_i^{-1,k}\right)$ for any $i$, $\boldsymbol{c}^{-1} = \frac{1}{N} \sum_i \boldsymbol{c}_i^{-1}$, momentum $\boldsymbol{g}^{-1} = \boldsymbol{c}^{-1}$, global learning rate $\gamma$, local learning rate $\eta$, and momentum parameter $\beta$
2: **for** $t = 0, \cdots, T-1$ **do**
3:     **Central Server:** Uniformly sample clients $\mathcal{S}_t \subseteq \{1, \cdots, N\}$ with $|\mathcal{S}_t| = S$
4:     **for** each client $i \in \mathcal{S}_t$ in parallel **do**
5:         Initialize local model $\boldsymbol{\theta}_i^{t,0} = \boldsymbol{\theta}^t$
6:         **for** $k = 0, \cdots, K-1$ **do**
7:             Compute $\boldsymbol{g}_i^{t,k} = \beta \left(\nabla F\left(\boldsymbol{\theta}_i^{t,k}; \boldsymbol{\xi}_i^{t,k}\right) - \boldsymbol{c}_i^{t-1} + \boldsymbol{c}^{t-1}\right) + (1-\beta)\boldsymbol{g}^{t-1}$
8:             Update local model $\boldsymbol{\theta}_i^{t,k+1} = \boldsymbol{\theta}_i^{t,k} - \eta \frac{\boldsymbol{g}_i^{t,k}}{\|\boldsymbol{g}_i^{t,k}\|}$
9:         **end for**
10:         Update control variate $\boldsymbol{c}_i^t = \frac{1}{K} \sum_{k=0}^{K-1} \nabla F\left(\boldsymbol{\theta}_i^{t,k}; \boldsymbol{\xi}_i^{t,k}\right)$ ( set $\boldsymbol{c}_i^t = \boldsymbol{c}_i^{t-1}$ for $i \notin \mathcal{S}_t$)
11:         Upload $\boldsymbol{\theta}_i^{t,K}$ and $\boldsymbol{c}_i^t$ to central server
12:     **end for**
        **Central server:**
13:     Aggregate local updates $\overline{\boldsymbol{g}}^t = \frac{1}{\eta S K} \sum_{i \in \mathcal{S}_t} \left(\boldsymbol{\theta}^t - \boldsymbol{\theta}_i^{t,K}\right)$
14:     Update global model $\boldsymbol{\theta}^{t+1} = \boldsymbol{\theta}^t - \gamma \overline{\boldsymbol{g}}^t$
15:     Aggregate control variate $\boldsymbol{c}^t = \boldsymbol{c}^{t-1} + \frac{1}{N} \sum_{i \in \mathcal{S}_t} \left(\boldsymbol{c}_i^t - \boldsymbol{c}_i^{t-1}\right)$
16:     Aggregate momentum $\boldsymbol{g}^t = \beta \left(\frac{1}{S} \sum_{i \in \mathcal{S}_t} \left(\boldsymbol{c}_i^t - \boldsymbol{c}_i^{t-1}\right) + \boldsymbol{c}^{t-1}\right) + (1-\beta)\boldsymbol{g}^{t-1}$
17:     Download $\boldsymbol{\theta}^{t+1}$, $\beta \boldsymbol{c}^t + (1-\beta)\boldsymbol{g}^t$ to all clients
18: **end for**

---

In this section, we propose PAdaMFed, a problem-parameter-agnostic algorithm for nonconvex FL based on adaptive stepsizes and client-side momentum. PAdaMFed is designed to operate independently of any problem-specific parameters, handle arbitrarily heterogeneous data, and accommodate partial client participation.

## 3.1 ALGORITHM DEVELOPMENT OF PADAMFED

Our approach builds upon the well-established SCAFFOLD algorithm (Karimireddy et al., 2020b), which was designed to address "client drift" in FL, where local models significantly deviate from the global model due to partial participation and data heterogeneity. The core concept of SCAFFOLD is the utilization of control variates to correct the drift between client updates and the global model. Specifically, the server maintains a global control variate, denoted $c^t$, to represent the average model update direction, while each client maintains a local control variate, denoted by $c_i^t$ for all $i \in \{1, \ldots, N\}$, to track individual update directions. Client updates are subsequently adjusted using the difference between local and global control variates. SCAFFOLD demonstrates faster and more stable convergence compared to the seminal FedAvg algorithm (McMahan et al., 2017).

In this paper, we extend the original SCAFFOLD framework by incorporating client-side momentum and local adaptive stepsizes, as outlined in Algorithm 1. Specifically, in Step 7 of Algorithm 1, the local descent direction of client $i$ at global round $t$ and local step $k$ is computed as:

$$g_i^{t,k} = \beta \left( \nabla F \left( \theta_i^{t,k}; \xi_i^{t,k} \right) - c_i^{t-1} + c^{t-1} \right) + (1 - \beta)g^{t-1}.$$

In this expression, the term $\nabla F \left( \theta_i^{t,k}; \xi_i^{t,k} \right)$ represents the stochastic gradient at the current local model $\theta_i^{t,k}$ with the sample $\xi_i^{t,k}$. The term $c^{t-1} - c_i^{t-1}$ adjusts the difference between the global and local control variates, helping to mitigate client drift. The term $g^{t-1}$ denotes the current global momentum, essential for stabilizing and accelerating the convergence across clients.

In Step 8 of Algorithm 1, the local model for each client $i$ is updated by: $\theta_i^{t,k+1} = \theta_i^{t,k} - \eta \frac{g_i^{t,k}}{\|g_i^{t,k}\|}$. Here, an adaptive stepsize $\eta/\|g_i^{t,k}\|$ is utilized by normalizing the descent direction vector $g_i^{t,k}$. This normalization guarantees that the progresses from all clients have a uniform magnitude, preventing the disproportionate impact of any individual client on the global model update. It also provides us the convenience on quantifying the distance between consecutive models in our theoretical analysis, maintaining that $\left\| \theta_i^{t,k+1} - \theta_i^{t,k} \right\| = \eta \left\| g_i^{t,k}/\|g_i^{t,k}\| \right\| = \eta$ for all $i, k, t$.

Additionally, since $c_i^t = \frac{1}{K} \sum_{k=0}^{K-1} \nabla F \left( \theta_i^{t,k}; \xi_i^{t,k} \right)$ for all $i$, the momentum update in Step 16 of Algorithm 1 can be expressed as:

$$g^t = \beta \left( \frac{1}{S} \sum_{i \in \mathcal{S}_t} \left( \frac{1}{K} \sum_{k=0}^{K-1} \nabla F \left( \theta_i^{t,k}; \xi_i^{t,k} \right) - c_i^{t-1} \right) + c^{t-1} \right) + (1 - \beta)g^{t-1}. \quad (2)$$

This equation accumulates the descent directions across clients and iterations that we have $g^t = \sum_{i,k} g_i^{t,k}$. With $g^t$, the optimization trajectories at each client are smoothed by the descent directions of other clients, enhancing the robustness of the optimization process against variability in local updates caused by data heterogeneity. Notably, our PAdaMFed algorithm maintains the communication workload of the SCAFFOLD for both uplink and downlink.

## 3.2 ACCELERATING PADAMFED WITH VARIANCE REDUCTION

Variance reduction is an effective technique to accelerate convergence and enhance the stability of FL, particularly when dealing with heterogeneous data and limited client participation. In this subsection, we enhance PAdaMFed by integrating a variance reduction component into each client's descent direction, resulting in our PAdaMFed-VR algorithm. The detailed procedures of PAdaMFed-VR are provided in Appendix B.

PAdaMFed-VR differs from PAdaMFed primarily in its computation of the local gradient. Specifically, Step 7 of Algorithm 1 is replaced with:

$$g_i^{t,k} = \nabla F \left( \theta_i^{t,k}; \xi_i^{t,k} \right) + \beta \left( c^{t-1} - c_i^{t-1} \right) + (1 - \beta) \left( g^{t-1} - \nabla F \left( \theta^{t-1}; \xi_i^{t,k} \right) \right),$$

where $\nabla F \left( \theta^{t-1}; \xi_i^{t,k} \right)$ represents the variance reduction component. Our variance reduction design follows the principle of STORM (Cutkosky & Orabona, 2019) to make more efficient sample utilization. For each local update, the sample $\xi_i^{t,k}$ is used twice: 1) to compute the gradient based on

the current local model $\boldsymbol{\theta}_i^{t,k}$; and 2) to evaluate the gradient at the previous global model $\boldsymbol{\theta}^{t-1}$. This dual usage of each sample mitigates the influence of within-client gradient noise, enabling more accurate estimation of gradient directions.

# 4 THEORETICAL RESULTS AND COMPARISONS WITH PRIOR WORK

## 4.1 THEORETICAL RESULTS

**Theorem 1.** *Suppose that Assumptions 2 and 3 hold. Let the local and global learning rates of PAdaMFed be $\eta = \frac{1}{K\sqrt{T}}$ and $\gamma = \frac{(SK)^{1/4}}{T^{3/4}}$, respectively, the momentum parameter be $\beta = \sqrt{\frac{SK}{T}}$, and $\{\boldsymbol{\theta}^t\}_{t\geq0}$ be the iterates generated by Algorithm 1. Then, it holds for all $T \geq 1$ that*

$$\frac{1}{T}\sum_{t=0}^{T-1}\mathbb{E}\left\|\nabla f\left(\boldsymbol{\theta}^t\right)\right\| \leq \mathcal{O}\left(\frac{\Delta + L + \sigma + \sqrt{L}\sigma}{(SKT)^{\frac{1}{4}}} + \frac{\sqrt{SK}\sigma + L}{\sqrt{T}}\right),$$

*where $\Delta := f\left(\boldsymbol{\theta}^0\right) - \min_{\boldsymbol{\theta}} f(\boldsymbol{\theta})$.*

**Theorem 2.** *Suppose that Assumptions 1 and 3 hold. Let the local and global learning rates of PAdaMFed-VR be $\eta = \frac{1}{KT}$ and $\gamma = \frac{(SK)^{1/3}}{T^{2/3}}$, respectively, the momentum parameter be $\beta = \frac{(SK)^{1/3}}{T^{2/3}}$, and $\{\boldsymbol{\theta}^t\}_{t\geq0}$ be the iterates generated by Algorithm 2. Then, it holds for all $T \geq 1$ that*

$$\frac{1}{T}\sum_{t=0}^{T-1}\mathbb{E}\left\|\nabla f\left(\boldsymbol{\theta}^t\right)\right\| \leq \mathcal{O}\left(\frac{\Delta + L + \sigma}{(SKT)^{\frac{1}{3}}} + \frac{(L+\sigma)(SK)^{\frac{1}{3}}}{T^{\frac{2}{3}}}\right).$$

**Remark 1.** *According to Theorem 1, PAdaMFed converges to an $\epsilon$-stationary point [1] in expectation within $\mathcal{O}\left(\frac{1}{SK\epsilon^4}\right)$ communication rounds. This convergence rate is improved to $\mathcal{O}\left(\frac{1}{SK\epsilon^3}\right)$ when incorporating variance reduction, as shown in Theorem 2. Furthermore, both algorithms demonstrate linear speedup with respect to the number of participating clients $S$ and local update steps $K$.*

**Remark 2.** *In PAdaMFed, setting $SK = \mathcal{O}(T^{\frac{1}{3}})$ yields a sample complexity[2] of $\mathcal{O}(\epsilon^{-4})$ with communication complexity $\mathcal{O}(\epsilon^{-3})$. Similarly, by setting $SK = \mathcal{O}(\sqrt{T})$, PAdaMFed-VR achieves the best-known sample complexity of $\mathcal{O}(\epsilon^{-3})$ with a communication complexity of $\mathcal{O}(\epsilon^{-2})$ to find an $\epsilon$-stationary point (Wu et al., 2023).*

**Remark 3.** *In traditional FL, selecting optimal stepsizes theoretically requires knowledge of problem-specific parameters, which are often unavailable. Consequently, in real-world FL scenarios, stepsizes must be tuned empirically—a process that is labor-intensive, time-consuming, and sometimes even impractical. In our PAdaMFed and PAdaMFed-VR, the stepsizes are explicitly determined by system-defined constants (the numbers of participating clients $S$, local update steps $K$, and communication rounds $T$), without requiring any problem-specific parameters. This hyperparameter-independent nature simplifies implementation, enhances robustness, and facilitates the deployment of our algorithm across diverse FL applications.*

## 4.2 PROOF SKETCH

Our theoretical proof starts from the $L$-smoothness property of the loss function $f(\boldsymbol{\theta})$, which yields the following inequality:

$$f\left(\boldsymbol{\theta}^{t+1}\right) - f\left(\boldsymbol{\theta}^t\right) \leq 2\gamma\left\|\nabla f\left(\boldsymbol{\theta}^t\right) - \boldsymbol{g}^t\right\| - \gamma\left\|\nabla f\left(\boldsymbol{\theta}^t\right)\right\| + \frac{\gamma}{SK}\sum_{i\in\mathcal{S}_{t,k}}\left\|\boldsymbol{g}_i^{t,k} - \boldsymbol{g}^t\right\| + \frac{\gamma^2 L}{2}.$$

---

[1] A point $\boldsymbol{\theta}$ is said to be $\epsilon$-stationary if $\|\nabla f\left(\boldsymbol{\theta}\right)\| \leq \epsilon$. Note that for any $\epsilon$-stationary point defined using $\|\nabla f(\boldsymbol{\theta})\|^2$, one can derive the corresponding guarantee for $\|\nabla f(\boldsymbol{\theta})\|$ based on the following relationship:

$$\frac{1}{T}\sum_{t=1}^{T-1}\mathbb{E}\|\nabla f(\boldsymbol{\theta}^t)\| = \frac{1}{T}\sum_{t=1}^{T-1}\mathbb{E}\sqrt{\|\nabla f(\boldsymbol{\theta}^t)\|^2} \leq \frac{1}{T}\sum_{t=1}^{T-1}\sqrt{\mathbb{E}\|\nabla f(\boldsymbol{\theta}^t)\|^2} \leq \sqrt{\frac{1}{T}\sum_{t=1}^{T-1}\mathbb{E}\|\nabla f(\boldsymbol{\theta}^t)\|^2}$$

where the first and second inequalities utilize Jensen's inequality as the square root function is concave. Therefore, we can align our results with the metric of $\frac{1}{T}\sum_{t=1}^{T}\|\nabla f(\boldsymbol{\theta}^t)\|^2$ by taking square root on both sides of their convergence bounds.

[2] Total number of samples across clients, local updates, and communication.

To establish an upper bound for $\frac{1}{T} \sum_{t=0}^{T-1} \mathbb{E} \|\nabla f (\boldsymbol{\theta}^t)\|$, we must quantify two key terms: $\frac{1}{T} \sum_{t=0}^{T-1} \mathbb{E} \|\nabla f (\boldsymbol{\theta}^t) - \boldsymbol{g}^t\|$ and $\frac{1}{T} \sum_{t=0}^{T-1} \frac{1}{SK} \mathbb{E} \left[ \sum_{i \in \mathcal{S}_{t,k}} \left\| \boldsymbol{g}_i^{t,k} - \boldsymbol{g}^t \right\| \right]$.

The momentum $\boldsymbol{g}^t$ is a recursive variable that accumulates values from previous rounds. By plugging into the expression of $\boldsymbol{g}^t$ (provided in (2)) into $\|\nabla f (\boldsymbol{\theta}^t) - \boldsymbol{g}^t\|$ and introducing the auxiliary term $f (\boldsymbol{\theta}^{t-1})$, we can recursively express $\|\nabla f (\boldsymbol{\theta}^t) - \boldsymbol{g}^t\|$ by its predecessor $\left\| \nabla f (\boldsymbol{\theta}^{t-1}) - \boldsymbol{g}^{t-1} \right\|$, scaled by a contraction coefficient $(1 - \beta)$. This substitution also introduces additional terms associated with stochastic gradients and control variates. Through meticulous control of all intermediate terms, we prove that $\frac{1}{T} \sum_{t=0}^{T-1} \mathbb{E} \|\nabla f (\boldsymbol{\theta}^t) - \boldsymbol{g}^t\|$ is upper bounded by $\mathcal{O}((SKT)^{-\frac{1}{4}})$ for PAdaMFed and by $\mathcal{O}((SKT)^{-\frac{1}{3}})$ for PAdaMFed-VR.

The term $\mathbb{E} \left[ \sum_{i \in \mathcal{S}_{t,k}} \left\| \boldsymbol{g}_i^{t,k} - \boldsymbol{g}^t \right\| \right]$ represents gradient dissimilarity across clients. While the heterogeneity bound (1) could readily control this term, our objective is to eliminate dependence on such bounds. Instead, we relax this term to $\mathbb{E} \left\| \nabla f_i (\boldsymbol{\theta}^{t-1}) - \boldsymbol{c}_i^{t-1} \right\|$, along with other controllable terms, by substituting the expressions for $\boldsymbol{g}_i^{t,k}$ and $\boldsymbol{g}^t$. Returning to our treatment of $\mathbb{E} \|\nabla f (\boldsymbol{\theta}^t) - \boldsymbol{g}^t\|$, we exploit the recursive property of the control variate $\boldsymbol{c}_i^{t-1}$ to bound $\mathbb{E} \left\| \nabla f_i (\boldsymbol{\theta}^{t-1}) - \boldsymbol{c}_i^{t-1} \right\|$. This, in turn, allows us to establish a bound for $\mathbb{E} \left[ \sum_{i \in \mathcal{S}_{t,k}} \left\| \boldsymbol{g}_i^{t,k} - \boldsymbol{g}^t \right\| \right]$.

Combining the above processes leads to our analytical results. For comprehensive proofs, please refer to Appendix A for Theorem 1 and Appendix B for Theorem 2.

**Intuition on the Algorithmic Features:** The efficacy of PAdaMFed stems from the synergistic integration of three indispensable components: local gradient normalization, client-side momentum, and control variates. 1) Gradient normalization serves as an adaptive learning rate scheme, automatically adjusting stepsizes based on the local optimization landscape. This design automatically allows larger steps in regions with small gradients (where more aggressive exploration is beneficial) and smaller steps in steep regions (where careful progress is needed). 2) Client-side momentum helps accelerate convergence while maintaining stability. First, it helps overcome local irregularities in the loss landscape by accumulating gradients over clients and iterations. Second, it accelerates progress in directions of consistent gradient agreement. 3) Furthermore, control variates align local updates with the global objective, reducing variance in gradient estimates and ensuring more consistent updates across heterogeneous client data. Collectively, these techniques ensure that the stepsizes are independent of problem-specific parameters such as smoothness constants and data heterogeneity bounds, simplifying the tuning process and enhancing robustness across diverse FL environments. Notably, our algorithms also eliminate the requirement of data heterogeneity bounds, further broadening their applicability.

### 4.3 COMPARISONS WITH PRIOR WORK

We compare PAdaMFed and PAdaMFed-VR with several representative algorithms for solving FL problems with heterogeneous data, as listed in Table 1.

**Comparisons of PAdaMFed with Prior FL Algorithms:** The SCAFFOLD (Karimireddy et al., 2020b) algorithm requires the smoothness parameter $L$ for stepsize tuning. However, its communication complexity, given by $\mathcal{O} \left( \left( \frac{K}{S} \right)^{\frac{1}{3}} \frac{L}{K \epsilon^4} \right)$, is suboptimal. MIME (Karimireddy et al., 2020a) improves this complexity to $\mathcal{O} \left( \frac{1}{SK \epsilon^4} \right)$ by incorporating server-level momentum. Nevertheless, MIME requires large-batch local gradients per round, and its learning rates depend on multiple problem parameters, including initial optimality gap $\Delta$ and heterogeneity bound $\sigma_h^2$, which are challenging to estimate.

FedSPS (Sohom Mukherjee, 2024) incorporates stochastic Polyak step-sizes into local client updates, achieving a communication complexity of $\mathcal{O} \left( \frac{1}{NK \epsilon^4} \right)$ in full participation scenarios. However, its analysis relies on the assumption of bounded data heterogeneity. Moreover, its stepsize tuning requires knowledge of the lower bounds of all loss functions, i.e., $\ell_i^*$ for all $i$, in addition to the smoothness parameter $L$, leading to significant problem-parameter dependence.

SCAFFOLD-M (Cheng et al., 2024) employs similar client-side momentum as PAdaMFed, removing the need for bounded data heterogeneity. However, SCAFFOLD-M's stepsizes depend on sev-

Table 1: Comparisons of algorithms for solving FL problems with heterogeneous data. (Shorthand notation: Add. Assump. = Additional assumptions aside from Assumptions 1–3, BDH = Bounded data heterogeneity define in (1), BG = Bounded gradient that $\|\nabla f_i(\boldsymbol{\theta})\| \leq G, \ \forall i, \boldsymbol{\theta}$, BHD = Bounded hessian dissimilarity that $\left\|\nabla^2 f_i(\boldsymbol{\theta}) - \nabla^2 f(\boldsymbol{\theta})\right\|^2 \leq \delta, \ \forall i, \boldsymbol{\theta}$)

| Algorithms | Add. Assump. | Stepsize Restrictions | Stepsize-Related Problem-Parameters | Communication Complexity |
|---|---|---|---|---|
| SCAFFOLD (Karimireddy et al., 2020b) | – | $\gamma = \sqrt{S}, \eta \leq \frac{1}{24\gamma KL}\left(\frac{S}{N}\right)^{\frac{2}{3}}$ | $L$ | $\mathcal{O}\left(\left(\frac{N}{S}\right)^{\frac{1}{3}}\frac{L}{K\epsilon^4}\right)$ |
| MIme (Karimireddy et al., 2020a) | BDH, BHD | $\eta = \sqrt{\frac{\Delta S}{L\tilde{G}TK^2}}, \tilde{G} = \sigma_h^2 + \frac{\sigma^2}{K}$ | $L, \Delta, \sigma^2, \sigma_h^2$ | $\mathcal{O}\left(\frac{1}{SK\epsilon^4}\right)$ |
| FedSPS Sohom Mukherjee (2024) | BDH | $\eta_i^{t,k} = \min\left\{\frac{F\left(\boldsymbol{\theta}_i^{t,k};\boldsymbol{\xi}_i^{t,k}\right)-\ell_i^*}{c\left\|\nabla F\left(\boldsymbol{\theta}_i^{t,k};\boldsymbol{\xi}_i^{t,k}\right)\right\|^2}, \eta_b\right\}$[1] $\eta_b \leq \min\left\{\frac{1}{2cL}, \frac{1}{25LK}\right\}$ | $L, \ell_i^*, \forall i$ | $\mathcal{O}\left(\frac{1}{NK\epsilon^4}\right)$ |
| SCAFFOLD-M (Cheng et al., 2024) | – | $\beta = \min\left\{1, \frac{S}{N^{\frac{2}{3}}}, \sqrt{\frac{L\Delta SK}{\sigma^2 T}}, \sqrt{\frac{L\Delta S^2}{G_0 N}}\right\}$[2] $\gamma = \frac{\beta}{L}, \eta KL \lesssim \min\left\{\frac{1}{\sqrt{S}}, \frac{1}{\beta K^{\frac{1}{4}}}, \frac{\sqrt{S}}{N}\right\}$ | $L, \Delta, \sigma^2, G_0$ | $\mathcal{O}\left(\frac{1}{SK\epsilon^4}\right)$ |
| PAdaMFed (This paper) | – | $\beta = \sqrt{\frac{SK}{T}}, \gamma = \frac{(SK)^{\frac{1}{4}}}{T^{\frac{3}{4}}}, \eta = \frac{1}{K\sqrt{T}}$ | – | $\mathcal{O}\left(\frac{1}{SK\epsilon^4}\right)$ |
| **Variance Reduction** | | | | |
| FAFED (Wu et al., 2023) | BDH, BG | $\eta_t \propto \frac{N^{\frac{2}{3}}}{Lt^{\frac{1}{3}}}, \beta_t \propto \eta_t^2$ | $L$ | $\mathcal{O}\left(\frac{1}{SK\epsilon^3}\right)$ |
| SCAFFOLD-M-VR (Cheng et al., 2024) | – | $\beta = \min\left\{\frac{S}{N}, \left(\frac{KL\Delta}{\sigma^2 T}\right)^{\frac{2}{3}}, S^{\frac{1}{3}}\right\}$ $\gamma L = \min\left\{1, \sqrt{\beta S}\right\}$ $\eta KL \lesssim \min\left\{\sqrt{\frac{\beta}{S}}, \left(\frac{\beta}{SK}\right)^{\frac{1}{4}}\right\}$ | $L, \Delta, \sigma^2$ | $\mathcal{O}\left(\frac{1}{S\sqrt{K}\epsilon^4}\right)$ |
| PAdaMFed-VR (This paper) | – | $\beta = \frac{(SK)^{\frac{1}{3}}}{T^{\frac{2}{3}}}, \gamma = \frac{(SK)^{\frac{1}{3}}}{T^{\frac{2}{3}}}, \eta = \frac{1}{KT}$ | – | $\mathcal{O}\left(\frac{1}{SK\epsilon^3}\right)$ |

[1] $\ell_i^* \leq \inf_{\boldsymbol{\xi}_i \in \mathcal{D}_i, \boldsymbol{\theta}} F(\boldsymbol{\theta}; \boldsymbol{\xi}_i)$ for any $i$, and $c$ is a constant to balance adaptivity and accuracy.
[2] $G_0 := \frac{1}{N}\sum_{i=1}^N \left\|\nabla f_i(\boldsymbol{\theta}^0)\right\|^2$.

eral problem-specific parameters, including the smoothness parameter $L$, initial optimal gap $\Delta$, and stochastic gradient variance $\sigma^2$, resulting in laborious stepsize tuning. In contrast, PAdaMFed is completely independent of problem-specific parameters while achieving start-of-the-art communication complexity.

**Comparisons of PAdaMFed-VR with Prior Variance-Reduced FL Algorithms:** FAFED (Wu et al., 2023) employing a momentum-based variance reduction with an adaptive matrix, achieving the best-known $\mathcal{O}(\epsilon^{-3})$ sample complexity and $\mathcal{O}(\epsilon^{-2})$ communication complexity through the use of diminishing stepsizes. However, FAFED requires stringent assumptions of bounded gradients and bounded data heterogeneity. Moreover, its learning rates are subject to several complex constraints and rely on problem-parameter-based algorithm tuning. SCAFFOLD-M-VR (Cheng et al., 2024) is a variance-reduced SCAFFOLD-M algorithm and, similarly, requires careful step size tuning based on multiple problem-dependent parameters. Nevertheless, it fails to achieve the best-known complexity established in the literature.

## 5 NUMERICAL EXPERIMENTS

In this section, we present experiments on two real-world datasets: EMNIST (Cohen et al., 2017) and CIFAR-10 (Li et al., 2017). We evaluate the proposed algorithms against several baselines,

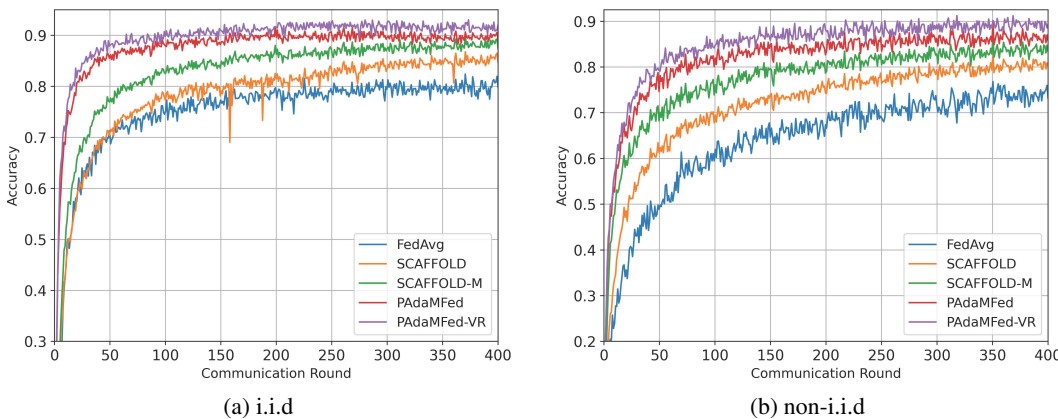

(a) i.i.d

(b) non-i.i.d

Figure 1: Test accuracy versus the number of communication rounds on the EMNIST dataset.

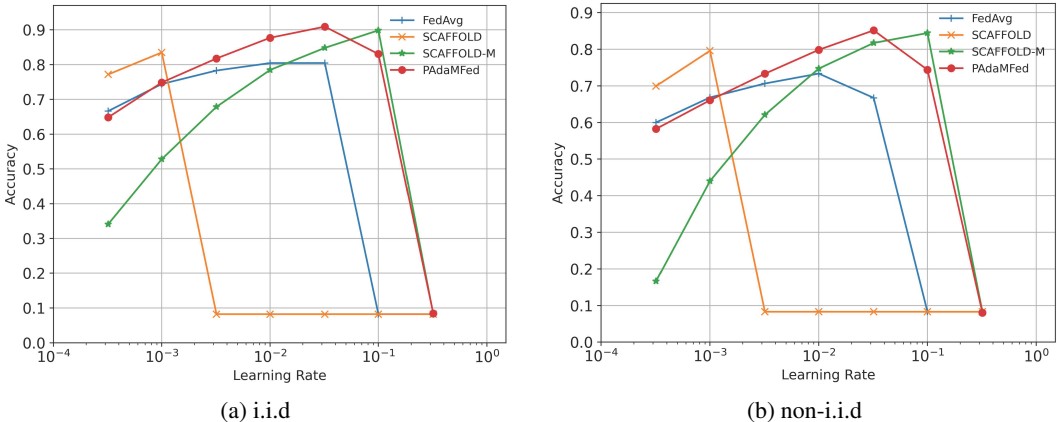

(a) i.i.d

(b) non-i.i.d

Figure 2: Test accuracy versus learning rate on the EMNIST dataset.

including FedAvg (McMahan et al., 2017), SCAFFOLD (Karimireddy et al., 2020b), SCAFFOLD-M (Cheng et al., 2024). Additionally, experiments are conducted under both i.i.d. and non-i.i.d. data distributions. Due to space limitations, the detailed experimental setup and additional simulation results are provided in Appendix C.

Figure 1 illustrates the test accuracy of various algorithms versus the number of communication rounds on the EMNIST dataset, with subfigure 1a representing i.i.d. data and subfigure 1b depicting non-i.i.d. data. The stepsizes for our algorithms, PAdaMFed and PAdaMFed-VR, are determined based on the theoretical guidance provided in Theorem 1 and Theorem 2, respectively. For fair comparisons, the hyperparameters of other algorithms in Figure 1 are optimized through grid search to achieve their best performance. The results demonstrate that our proposed methods significantly outperform all baseline algorithms—FedAvg, SCAFFOLD, and SCAFFOLD-M—in both convergence speed and test accuracy. Notably, although SCAFFOLD-M employs a similar momentum technique, it converges more slowly than PAdaMFed and achieves lower accuracy, validating the efficacy of our adaptive stepsize design. Building upon these advantages, the incorporation of variance reduction further enhances our methods' superiority through more efficient sample utilization. Moreover, the results on non-i.i.d. data in subfigure 1b demonstrate even greater performance margins than the i.i.d. case, highlighting the advantages of our algorithms.

Figure 2 compares the test accuracy of various algorithms versus the learning rate on the EMNIST dataset. All algorithms were evaluated over 400 communication rounds to ensure a fair comparison. We observe that our algorithm demonstrates superior robustness to stepsize selection, maintaining stable performance across a significantly wider range of learning rates compared to baseline methods. Specifically, it achieves test accuracy exceeding 0.8 across the stepsize range $[3 \times 10^{-3}, 10^{-1}]$

for i.i.d. data distributions (subfigure 2a) and above 0.7 for the same stepsize range under non-i.i.d. conditions (subfigure 2b). In contrast, baseline algorithms exhibit substantially narrower regions of stable performance, empirically validating our method's enhanced stepsize robustness.

## 6    CONCLUSIONS

This paper proposed a novel training approach, PAdaMFed, for nonconvex federated learning (FL) that eliminates dependency on problem-specific parameters and enables automatic generalization across diverse FL environments. PAdaMFed also removes the need for data heterogeneity bounds and accommodates partial client participation, further broadening its applicability. We provided rigorous theoretical analysis for PAdaMFed, demonstrating its state-of-the-art convergence under minimal assumptions, including the $L$-smoothness of loss functions and unbiased stochastic gradients with bounded within-client variance. Furthermore, we enhanced this convergence rate by incorporating variance reduction, achieving the best-known $\mathcal{O}(\epsilon^{-3})$ sampling complexity and $\mathcal{O}(\epsilon^{-2})$ communication complexity. Extensive numerical experiments demonstrated that our algorithms outperform existing representative FL approaches in both runtime efficiency and stepsize robustness.

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

## A    RELATED WORKS

**Adaptive Stepsize Methods:** Adaptive stepsize methods have gained significant attention in optimization literature due to their ability to automatically adjust learning rates based on the geometry of the objective function. These methods, such as Adam (Kingma & Ba, 2014), AdaGrad (Duchi et al., 2011), and RMSProp Hinton (2012), have demonstrated remarkable success in various machine learning tasks, particularly in handling sparse gradients and non-stationary objectives. In the context of FL, adaptive stepsizes offer several advantages. First, they eliminate the need for manual tuning of learning rates, which is especially beneficial in federated settings where global knowledge of the objective function's properties is limited. Second, adaptive methods can potentially mitigate the impact of data heterogeneity by allowing different update rates for different model parameters, effectively accounting for varying gradient statistics across clients.

FedOpt (Reddi et al., 2020) introduced adaptive optimization techniques like AdaGrad, Adam, and Yogi into FL, demonstrating improved convergence properties compared to the traditional FedAvg algorithm. Wang et al. (2020) proposed FedNova, which normalizes and scales local updates to mitigate objective inconsistency caused by partial client participation and data heterogeneity. FedBN (Li et al., 2021) employed local batch normalization to alleviate feature shift before averaging models, outperforming both classical FedAvg (McMahan et al., 2017) and FedProx (Li et al., 2020) for non-IID data. While these approaches demonstrated the advantages of adaptive methods for easing parameter tuning, none provides theoretical guarantees, and careful tuning of global learning rates remains essential.

**Momentum:** Momentum, on the other hand, is a technique that accelerates gradient descent by accumulating a velocity vector in directions of persistent reduction in the objective across iterations. In nonconvex optimization, momentum has been shown to help escape saddle points more efficiently and potentially reach better local optima (Cutkosky & Orabona, 2019). The incorporation of momentum in FL algorithms can provide several benefits. It can help smooth out the impact of heterogeneous updates from different clients, potentially leading to more stable and faster convergence. Moreover, momentum can aid in overcoming the challenges posed by partial client participation by maintaining a consistent optimization trajectory even when client participation varies across rounds.

Hsu et al. (2019) proposed FedAvgM, which adds a server-side momentum term to the FedAvg algorithm, demonstrating improved convergence rates and robustness to data heterogeneity. Wang et al. (2019) developed SlowMo, a momentum-based method using two nested loops to achieve faster convergence in distributed optimization. Both FedCM (Xu et al., 2021) and Cheng et al. (2024) investigated the integration of client-side momentum in FedAvg to effectively tackle client heterogeneity and partial participation in FL.

**Combination of Adaptive Stepsizes and Momentum:** Recent works have explored combining adaptive methods and momentum in FL. FedAMS (Chen et al., 2020) implements a local AMSGrad scheme for FL, demonstrating fast convergence with low communication cost. MIme (Karimireddy et al., 2020a) combines control variates with server-level momentum at every local update to mimic centralized methods running on IID data, outperforming centralized methods but requiring a large-batch local gradient per round for each client. Wu et al. (2023) introduced FAFED, a momentum-based variance reduction scheme integrated with an adaptive matrix, attaining the best-known sample and communication complexity when using diminishing stepsizes. While these methods demonstrate improved performance, they still require careful tuning of global learning rates.

A recent contribution, FedSPS (Sohom Mukherjee, 2024), claims to be the first fully locally adaptive method for FL with minimal hyperparameter tuning. While promising, this approach relies on stringent assumptions of bounded data heterogeneity and gradients. Moreover, it fails to converge to optima with constant stepsizes and requires adjustment of a maximum stepsize threshold based on the smoothness parameter, thus retaining some hyperparameter dependence. The limitations of existing approaches underscore the critical need for more robust, adaptive FL algorithms capable of operating effectively across diverse scenarios without extensive parameter tuning.

This paper makes a significant advancement by proposing a novel algorithm, PAdaMFed, that completely eliminates dependency on problem-specific parameters. All stepsizes in our approach are explicitly determined by the number of participating clients, local updates, and communication rounds.

To the best of our knowledge, this is the first algorithm that achieves such problem-parameter independence in FL. Moreover, our algorithm inherently manages data heterogeneity and partial client participation without requiring any heterogeneity bound among clients, which is also nontrivial.

Data heterogeneity has been extensively studied in FL. However, existing algorithms either depend on bounded data heterogeneity (e.g., FedAvg (McMahan et al., 2017), SCAFFOLD (Karimireddy et al., 2020b), FedProx (Li et al., 2020), and MIme (Karimireddy et al., 2020a)) or fall short of achieving state-of-the-art convergence rates (e.g., VRL-SGD (Liang et al., 2019) and LED (Alghunaim, 2024)). Recently, Cheng et al. (2024) demonstrated that momentum can eliminate the data heterogeneity constraint in the FedAvg and SCAFFOLD algorithms while achieving state-of-the-art convergence results. Wang et al. (2024) introduced DuDe-ASGD for asynchronous FL, which can effectively handle arbitrarily heterogeneous data by leveraging stale stochastic gradients. However, their algorithms require carefully designed stepsizes based on hyperparameters. In contrast, our algorithm accommodates arbitrary data heterogeneity while achieving complete problem-parameter independence.

A notable concurrent work by (Li et al., 2024) also explores problem-parameter-free algorithms in the context of decentralized non-convex optimization. While both studies target problem-parameter-free optimization, our work addresses the unique challenges inherent to federated learning settings. The fundamental distinction lies in our treatment of client drift—a critical phenomenon arising from multiple local updates and data heterogeneity in federated environments. Our key technical contributions beyond (Li et al., 2024) are twofold: 1) The development of a novel framework integrating control variates with momentum to mitigate client drift, requiring sophisticated theoretical analysis due to their complex interactions; 2) The successful elimination of explicit heterogeneity bounds, which were previously considered essential in federated learning literature.

## A  THEORETICAL ANALYSIS OF PADAMFED

Our analysis is based on the following useful lemmas.

**Lemma 1.** *For any $t$, we have*

$$\frac{1}{NK}\sum_{i,k}\left\|\boldsymbol{\theta}_i^{t,k}-\boldsymbol{\theta}^t\right\|^2 \leq \frac{1}{3}\eta^2 K^2 \quad and \quad \frac{1}{NK}\sum_{i,k}\left\|\boldsymbol{\theta}_i^{t,k}-\boldsymbol{\theta}^t\right\| \leq \frac{1}{2}\eta K.$$

*Proof.* From the update rule of local model, for any $i,k$ and $t$, we have

$$\left\|\boldsymbol{\theta}_i^{t,k+1}-\boldsymbol{\theta}_i^{t,k}\right\| = \eta\left\|\frac{\boldsymbol{g}_i^{t,k}}{\left\|\boldsymbol{g}_i^{t,k}\right\|}\right\| \leq \eta.$$

Then,

$$\left\|\boldsymbol{\theta}_i^{t,k}-\boldsymbol{\theta}^t\right\|^2 = \left\|\sum_{j=0}^{k-1}\left(\boldsymbol{\theta}_i^{t,j+1}-\boldsymbol{\theta}_i^{t,j}\right)\right\|^2 \leq k\sum_{j=0}^{k-1}\left\|\boldsymbol{\theta}_i^{t,j+1}-\boldsymbol{\theta}_i^{t,j}\right\|^2 \leq \eta^2 k^2.$$

Summing the above inequality over $i$ and $k$ yields

$$\frac{1}{NK}\sum_{i,k}\left\|\boldsymbol{\theta}_i^{t,k}-\boldsymbol{\theta}^t\right\|^2 \leq \frac{\eta^2}{K}\sum_{k=0}^{K-1}k^2 \leq \frac{\eta^2}{6K}(K-1)K(2K-1) \leq \frac{1}{3}\eta^2 K^2.$$

Similarly, we have

$$\frac{1}{NK}\sum_{i,k}\left\|\boldsymbol{\theta}_i^{t,k}-\boldsymbol{\theta}^t\right\| = \frac{1}{NK}\sum_{i,k}\left(\left\|\boldsymbol{\theta}_i^{t,k}-\boldsymbol{\theta}^t\right\|^2\right)^{\frac{1}{2}} \leq \frac{\eta}{K}\sum_{k=0}^{K-1}k \leq \frac{1}{2}\eta K.$$

$\square$

These inequalities in Lemma 1 are frequently used in our analysis.

**Lemma 2.** *Given vectors $\boldsymbol{\omega}_1, \cdots, \boldsymbol{\omega}_N \in \mathbb{R}^d$ and $\overline{\boldsymbol{\omega}} = \frac{1}{N} \sum_{i=1}^{N} \boldsymbol{\omega}_i$, if we sample $\mathcal{S} \subset \{1, \cdots, N\}$ uniformly randomly such that $|\mathcal{S}| = S$, then it holds that*

$$\mathbb{E}\left[\left\|\frac{1}{S}\sum_{i\in\mathcal{S}}\boldsymbol{\omega}_i\right\|^2\right] \le \|\overline{\boldsymbol{\omega}}\|^2 + \frac{1}{SN}\sum_{i=1}^{N}\|\boldsymbol{\omega}_i - \overline{\boldsymbol{\omega}}\|^2.$$

*Proof.* Letting $\mathbb{1}\{i \in \mathcal{S}\}$ be the indicator for the event $i \in \mathcal{S}$, we prove this lemma by direct calculation as follows:

$$\mathbb{E}\left[\left\|\frac{1}{S}\sum_{i\in\mathcal{S}}\boldsymbol{\omega}_i\right\|^2\right] = \mathbb{E}\left[\left\|\frac{1}{S}\sum_{i=1}^{N}\boldsymbol{\omega}_i\mathbb{1}\{i\in\mathcal{S}\}\right\|^2\right]$$

$$= \frac{1}{S^2}\mathbb{E}\left[\left(\sum_i\|\boldsymbol{\omega}_i\|^2\mathbb{1}\{i\in\mathcal{S}\} + 2\sum_{i<j}\boldsymbol{\omega}_i^\top\boldsymbol{\omega}_j\mathbb{1}\{i,j\in\mathcal{S}\}\right)\right]$$

$$= \frac{1}{SN}\sum_{i=1}^{N}\|\boldsymbol{\omega}_i\|^2 + \frac{1}{S^2}\frac{S(S-1)}{N(N-1)}2\sum_{i<j}\boldsymbol{\omega}_i^\top\boldsymbol{\omega}_j$$

$$= \frac{1}{SN}\sum_{i=1}^{N}\|\boldsymbol{\omega}_i\|^2 + \frac{1}{S^2}\frac{S(S-1)}{N(N-1)}\left(\left\|\sum_{i=1}^{N}\boldsymbol{\omega}_i\right\|^2 - \sum_{i=1}^{N}\|\boldsymbol{\omega}_i\|^2\right)$$

$$= \frac{N-S}{S(N-1)}\frac{1}{N}\sum_{i=1}^{N}\|\boldsymbol{\omega}_i\|^2 + \frac{N(S-1)}{S(N-1)}\|\overline{\boldsymbol{\omega}}\|^2$$

$$= \frac{N-S}{S(N-1)}\frac{1}{N}\sum_{i=1}^{N}\|\boldsymbol{\omega}_i - \overline{\boldsymbol{\omega}}\|^2 + \|\overline{\boldsymbol{\omega}}\|^2$$

$$\le \frac{1}{SN}\sum_{i=1}^{N}\|\boldsymbol{\omega}_i - \overline{\boldsymbol{\omega}}\|^2 + \|\overline{\boldsymbol{\omega}}\|^2,$$

where the last inequality uses the fact that $\frac{N-S}{N-1} \le 1$ for any nonempty set $\mathcal{S}$. $\square$

From the $L$-smoothness of $f(\cdot)$ in Assumption 2, we have

$$f\left(\boldsymbol{\theta}^{t+1}\right) - f\left(\boldsymbol{\theta}^t\right)$$

$$\le \nabla f\left(\boldsymbol{\theta}^t\right)^\top\left(\boldsymbol{\theta}^{t+1} - \boldsymbol{\theta}^t\right) + \frac{L}{2}\left\|\boldsymbol{\theta}^{t+1} - \boldsymbol{\theta}^t\right\|^2$$

$$\overset{(a)}{\le} -\gamma\nabla f\left(\boldsymbol{\theta}^t\right)^\top\left(\frac{1}{SK}\sum_{i\in\mathcal{S}_{t,k}}\frac{\boldsymbol{g}_i^{t,k}}{\left\|\boldsymbol{g}_i^{t,k}\right\|}\right) + \frac{\gamma^2 L}{2}$$

$$= -\gamma\left(\nabla f\left(\boldsymbol{\theta}^t\right) - \boldsymbol{g}^t\right)^\top\left(\frac{1}{SK}\sum_{i\in\mathcal{S}_{t,k}}\frac{\boldsymbol{g}_i^{t,k}}{\left\|\boldsymbol{g}_i^{t,k}\right\|}\right) - \gamma\left(\boldsymbol{g}^t\right)^\top\left(\frac{1}{SK}\sum_{i\in\mathcal{S}_{t,k}}\frac{\boldsymbol{g}_i^{t,k}}{\left\|\boldsymbol{g}_i^{t,k}\right\|}\right) + \frac{\gamma^2 L}{2}$$

$$\le \gamma\left\|\nabla f\left(\boldsymbol{\theta}^t\right) - \boldsymbol{g}^t\right\| - \gamma\left(\boldsymbol{g}^t\right)^\top\left(\frac{1}{SK}\sum_{i\in\mathcal{S}_{t,k}}\frac{\boldsymbol{g}_i^{t,k}}{\left\|\boldsymbol{g}_i^{t,k}\right\|} - \frac{\boldsymbol{g}^t}{\|\boldsymbol{g}^t\|}\right) - \gamma\left\|\boldsymbol{g}^t\right\| + \frac{\gamma^2 L}{2}$$

$$\overset{(b)}{\le} 2\gamma\left\|\nabla f\left(\boldsymbol{\theta}^t\right) - \boldsymbol{g}^t\right\| - \gamma\left\|\nabla f\left(\boldsymbol{\theta}^t\right)\right\| + \gamma\left\|\boldsymbol{g}^t\right\|\left\|\frac{1}{SK}\sum_{i\in\mathcal{S}_{t,k}}\frac{\boldsymbol{g}_i^{t,k}}{\left\|\boldsymbol{g}_i^{t,k}\right\|} - \frac{\boldsymbol{g}^t}{\|\boldsymbol{g}^t\|}\right\| + \frac{\gamma^2 L}{2}$$

$$\overset{(c)}{\le} 2\gamma\left\|\nabla f\left(\boldsymbol{\theta}^t\right) - \boldsymbol{g}^t\right\| - \gamma\left\|\nabla f\left(\boldsymbol{\theta}^t\right)\right\| + \frac{\gamma}{SK}\sum_{i\in\mathcal{S}_{t,k}}\left\|\boldsymbol{g}_i^{t,k} - \boldsymbol{g}^t\right\| + \frac{\gamma^2 L}{2}, \qquad (A1)$$

where $(a)$ uses the inequality that $\left\|\boldsymbol{\theta}^{t+1} - \boldsymbol{\theta}^t\right\| = \left\|\frac{\gamma}{SK}\sum_{i\in\mathcal{S}_t,k}\frac{\boldsymbol{g}_i^{t,k}}{\left\|\boldsymbol{g}_i^{t,k}\right\|}\right\| \leq \gamma$, $(b)$ is based on $\gamma\left\|\nabla f\left(\boldsymbol{\theta}^t\right)\right\| - \gamma\left\|\boldsymbol{g}^t\right\| \leq \gamma\left\|\nabla f\left(\boldsymbol{\theta}^t\right) - \boldsymbol{g}^t\right\|$ and $(c)$ is from the following relation:

$$
\begin{aligned}
\left\|\boldsymbol{g}^t\right\|\left\|\frac{1}{SK}\sum_{i\in\mathcal{S}_t,k}\frac{\boldsymbol{g}_i^{t,k}}{\left\|\boldsymbol{g}_i^{t,k}\right\|} - \frac{\boldsymbol{g}^t}{\left\|\boldsymbol{g}^t\right\|}\right\| &= \frac{\left\|\boldsymbol{g}^t\right\|}{SK}\left\|\sum_{i\in\mathcal{S}_t,k}\left(\frac{\boldsymbol{g}_i^{t,k}}{\left\|\boldsymbol{g}_i^{t,k}\right\|} - \frac{\boldsymbol{g}_i^{t,k}}{\left\|\boldsymbol{g}^t\right\|}\right)\right\| \\
&= \frac{\left\|\boldsymbol{g}^t\right\|}{SK}\left\|\sum_{i\in\mathcal{S}_t,k}\frac{\left\|\boldsymbol{g}^t\right\| - \left\|\boldsymbol{g}_i^{t,k}\right\|}{\left\|\boldsymbol{g}^t\right\|\left\|\boldsymbol{g}_i^{t,k}\right\|}\boldsymbol{g}_i^{t,k}\right\| \\
&\leq \frac{\left\|\boldsymbol{g}^t\right\|}{SK}\sum_{i\in\mathcal{S}_t,k}\frac{\left|\left\|\boldsymbol{g}^t\right\| - \left\|\boldsymbol{g}_i^{t,k}\right\|\right|}{\left\|\boldsymbol{g}^t\right\|\left\|\boldsymbol{g}_i^{t,k}\right\|}\left\|\boldsymbol{g}_i^{t,k}\right\| \\
&= \frac{1}{SK}\sum_{i\in\mathcal{S}_t,k}\left|\left\|\boldsymbol{g}^t\right\| - \left\|\boldsymbol{g}_i^{t,k}\right\|\right| \\
&\leq \frac{1}{SK}\sum_{i\in\mathcal{S}_t,k}\left\|\boldsymbol{g}_i^{t,k} - \boldsymbol{g}^t\right\|.
\end{aligned}
$$

Taking expectation on both sides of (A1), we obtain

$$
\begin{aligned}
\gamma\mathbb{E}\left\|\nabla f\left(\boldsymbol{\theta}^t\right)\right\| \leq& \mathbb{E}\left[f\left(\boldsymbol{\theta}^t\right) - f\left(\boldsymbol{\theta}^{t+1}\right)\right] + 2\gamma\mathbb{E}\left\|\nabla f\left(\boldsymbol{\theta}^t\right) - \boldsymbol{g}^t\right\| \\
&+ \frac{\gamma}{SK}\mathbb{E}\left[\sum_{i\in\mathcal{S}_t,k}\left\|\boldsymbol{g}_i^{t,k} - \boldsymbol{g}^t\right\|\right] + \frac{\gamma^2 L}{2}.
\end{aligned}
$$

Summing the above inequality over $t$ and dividing it by $\gamma T$, we have

$$
\begin{aligned}
\frac{1}{T}\sum_{t=0}^{T-1}\mathbb{E}\left\|\nabla f\left(\boldsymbol{\theta}^t\right)\right\| \leq& \frac{1}{\gamma T}\mathbb{E}\left[f\left(\boldsymbol{\theta}^0\right) - f\left(\boldsymbol{\theta}^T\right)\right] + \frac{2}{T}\sum_{t=0}^{T-1}\mathbb{E}\left\|\nabla f\left(\boldsymbol{\theta}^t\right) - \boldsymbol{g}^t\right\| \\
&+ \frac{1}{SKT}\sum_{t=0}^{T-1}\mathbb{E}\left[\sum_{i\in\mathcal{S}_t}\left\|\boldsymbol{g}_i^{t,k} - \boldsymbol{g}^t\right\|\right] + \frac{\gamma L}{2}. \quad\quad\text{(A2)}
\end{aligned}
$$

We have the following results on the terms $\frac{1}{T}\sum_{t=0}^{T-1}\mathbb{E}\left\|\nabla f\left(\boldsymbol{\theta}^t\right) - \boldsymbol{g}^t\right\|$ and $\frac{1}{SKT}\sum_{t=0}^{T-1}\mathbb{E}\left[\sum_{i\in\mathcal{S}_t}\left\|\boldsymbol{g}_i^{t,k} - \boldsymbol{g}^t\right\|\right]$ in (A2).

**Lemma 3.** *Under Assumptions 2 and 3, the disparity* $\frac{1}{T}\sum_{t=0}^{T-1}\mathbb{E}\left\|\nabla f\left(\boldsymbol{\theta}^t\right) - \boldsymbol{g}^t\right\|$ *is upper bounded by:*

$$
\begin{aligned}
\frac{1}{T}\sum_{t=0}^{T-1}\mathbb{E}\left\|\nabla f\left(\boldsymbol{\theta}^t\right) - \boldsymbol{g}^t\right\| \leq& \frac{1}{\beta T}\left(\frac{1}{2}\eta\beta KL + \frac{3\sigma}{\sqrt{SK}}\right) + \frac{\gamma L}{\beta} + \sqrt{1 + \frac{10\beta}{S}}\eta KL + \sigma\sqrt{\frac{30\beta}{SK}} \\
&+ 2\gamma L\sqrt{\frac{\beta}{S}\left(1 + \frac{4N^2}{S^2}\right)} + \sqrt{\frac{\eta\sqrt{K}L\sigma}{2\sqrt{S}}}.
\end{aligned}
$$

**Lemma 4.** *Under Assumptions 2 and 3, the gradient dissimilarity* $\frac{1}{SKT}\sum_{t=0}^{T-1}\mathbb{E}\left[\sum_{i\in\mathcal{S}_t}\left\|\boldsymbol{g}_i^{t,k} - \boldsymbol{g}^t\right\|\right]$ *is upper bounded by:*

$$
\begin{aligned}
\frac{1}{SKT}\sum_{t=1}^{T}\mathbb{E}\left[\sum_{i\in\mathcal{S}_t,k}\left\|\boldsymbol{g}_i^{t,k} - \boldsymbol{g}^t\right\|\right] \leq& 2\beta\left(\left(1 + \frac{2}{\sqrt{K}}\right)\sigma + 2\eta KL + \left(1 + \frac{2N}{S}\right)\gamma L\right) \\
&+ \frac{8N\beta}{ST}\left(\frac{\sqrt{2}\sigma}{\sqrt{K}} + \sqrt{\frac{2S}{3N}}\eta KL\right).
\end{aligned}
$$

The proof of Lemma 3 is presented in subsection A.1 and the proof of Lemma 4 is presented in subsection A.2.

Set $\beta = \frac{\beta_0}{\sqrt{T}}, \gamma = \frac{\gamma_0}{T^{\frac{3}{4}}}$ and $\eta = \frac{1}{K\sqrt{T}}$. From Lemma 3, we know that

$$\frac{1}{T} \sum_{t=0}^{T-1} \mathbb{E} \left\| \nabla f \left( \boldsymbol{\theta}^t \right) - \boldsymbol{g}^t \right\|$$

$$\leq \frac{1}{\beta_0 \sqrt{T}} \left( \frac{\beta_0 L}{2T} + \frac{3\sigma}{\sqrt{SK}} \right) + \frac{\gamma_0 L}{\beta_0 T^{\frac{1}{4}}} + \sqrt{1 + \frac{10\beta_0}{S\sqrt{T}}} \frac{L}{\sqrt{T}} + \sigma \sqrt{\frac{30\beta_0}{SK\sqrt{T}}}$$

$$+ \frac{2\gamma_0 L}{T} \sqrt{\frac{\beta_0}{S} \left( 1 + \frac{4N^2}{S^2} \right)} + \sqrt{\frac{L\sigma}{2\sqrt{SKT}}}$$

$$\lesssim \frac{3\sigma}{\beta_0 \sqrt{SKT}} + \frac{\gamma_0 L}{\beta_0 T^{\frac{1}{4}}} + \frac{L}{\sqrt{T}} + \sigma \sqrt{\frac{30\beta_0}{SK\sqrt{T}}} + \sqrt{\frac{L\sigma}{2\sqrt{SKT}}}. \tag{A3}$$

Similarly, from Lemma 4, we know that

$$\frac{1}{SKT} \sum_{t=0}^{T-1} \mathbb{E} \left[ \sum_{i \in \mathcal{S}_{t,k}} \left\| \boldsymbol{g}_i^{t,k} - \boldsymbol{g}^t \right\| \right]$$

$$\leq \frac{2\beta_0}{\sqrt{T}} \left( \left( 1 + \frac{2}{\sqrt{K}} \right) \sigma + \frac{2L}{\sqrt{T}} + \left( 1 + \frac{2N}{S} \right) \frac{\gamma_0 L}{T^{\frac{3}{4}}} \right) + \frac{8N\beta_0}{ST^{\frac{3}{2}}} \left( \frac{\sqrt{2}\sigma}{\sqrt{K}} + \sqrt{\frac{2S}{3N}} \frac{L}{\sqrt{T}} \right)$$

$$\lesssim \frac{2\beta_0 \sigma}{\sqrt{T}} + \frac{4\beta_0 \sigma}{\sqrt{KT}}. \tag{A4}$$

Define the initial optimality gap $\Delta := f \left( \boldsymbol{\theta}^0 \right) - f^*$. Then, $f \left( \boldsymbol{\theta}^0 \right) - f \left( \boldsymbol{\theta}^T \right) \leq f \left( \boldsymbol{\theta}^0 \right) - f^* = \Delta$. Plugging (A3) and (A4) into (A2), we have

$$\frac{1}{T} \sum_{t=0}^{T-1} \mathbb{E} \left\| \nabla f \left( \boldsymbol{\theta}^t \right) \right\| \lesssim \frac{\Delta}{\gamma_0 T^{\frac{1}{4}}} + \frac{6\sigma}{\beta_0 \sqrt{SKT}} + \frac{2\gamma_0 L}{\beta_0 T^{\frac{1}{4}}} + \frac{2L}{\sqrt{T}} + 2\sigma \sqrt{\frac{30\beta_0}{SK\sqrt{T}}}$$

$$+ \frac{\sqrt{2L\sigma}}{(SKT)^{\frac{1}{4}}} + \frac{2\beta_0 \sigma}{\sqrt{T}} + \frac{4\beta_0 \sigma}{\sqrt{KT}} + \frac{\gamma_0 L}{2T^{\frac{3}{4}}}.$$

Let $\gamma_0 = (SK)^{\frac{1}{4}}$ and $\beta_0 = \sqrt{SK}$. Then, we have

$$\frac{1}{T} \sum_{t=0}^{T-1} \mathbb{E} \left\| \nabla f \left( \boldsymbol{\theta}^t \right) \right\| \leq \mathcal{O} \left( \frac{\Delta + L + \sigma + \sqrt{L\sigma}}{(SKT)^{\frac{1}{4}}} + \frac{\sqrt{SK}\sigma + L}{\sqrt{T}} \right).$$

By setting $SK \leq \mathcal{O} \left( T^{\frac{1}{3}} \right)$, we have $\frac{\sqrt{SK}}{\sqrt{T}} \propto \mathcal{O} \left( (SKT)^{-\frac{1}{4}} \right)$ and thus

$$\frac{1}{T} \sum_{t=0}^{T-1} \mathbb{E} \left\| \nabla f \left( \boldsymbol{\theta}^t \right) \right\| \leq \mathcal{O} \left( \frac{\Delta + L + \sigma + \sqrt{L\sigma}}{(SKT)^{\frac{1}{4}}} \right).$$

## A.1  PROOF OF LEMMA 3

The proof of Lemma 3 utilizes the following result.

**Lemma 5.** *For any $i, t$, define $\phi_i^t := \mathbb{E} \left\| \nabla f_i \left( \boldsymbol{\theta}^t \right) - \boldsymbol{c}_i^t \right\|^2$. Under Assumptions 2 and 3, we have*

$$\phi_i^t \leq \left( \frac{2\sigma^2}{K} + \frac{2S}{3N} \eta^2 K^2 L^2 \right) \left( 1 - \frac{S}{4N} \right)^{2t} + 4 \left( \frac{N^2}{S^2} \gamma^2 L^2 + \frac{\sigma^2}{K} + \frac{1}{3} \eta^2 K^2 L^2 \right), \forall i.$$

*Proof.* Since for any $t$, the $S$ elements in $\mathcal{S}_t$ are uniformly sampled from $\{1, \cdots, N\}$, we have

$$\boldsymbol{c}_i^t = \begin{cases} \boldsymbol{c}_i^{t-1} & \text{with probability} 1 - \frac{S}{N} \\ \frac{1}{K} \sum_k \nabla F \left( \boldsymbol{\theta}_i^{t,k}; \boldsymbol{\xi}_i^{t,k} \right) & \text{with probability} \frac{S}{N}. \end{cases}$$

Using Young's inequality repeatedly, we have

$$
\begin{aligned}
\phi_i^t &= \left(1 - \frac{S}{N}\right) \mathbb{E}\left\|\nabla f_i\left(\boldsymbol{\theta}^t\right) - \boldsymbol{c}_i^{t-1}\right\|^2 + \frac{S}{N}\mathbb{E}\left\|\frac{1}{K}\sum_k\left(\nabla f_i\left(\boldsymbol{\theta}^t\right) - \nabla F\left(\boldsymbol{\theta}_i^{t,k};\boldsymbol{\xi}_i^{t,k}\right)\right)\right\|^2 \\
&\leq \left(1 - \frac{S}{N}\right)\mathbb{E}\left\|\nabla f_i\left(\boldsymbol{\theta}^t\right) \mp \nabla f_i\left(\boldsymbol{\theta}^{t-1}\right) - \boldsymbol{c}_i^{t-1}\right\|^2 + \frac{S}{N}\left(\frac{2\sigma^2}{K} + \frac{2L^2}{K}\sum_k\mathbb{E}\left\|\boldsymbol{\theta}_i^{t,k} - \boldsymbol{\theta}^t\right\|^2\right) \\
&\leq \left(1 - \frac{S}{N}\right)\mathbb{E}\left[\left(1 + \frac{S}{2N}\right)\phi_i^{t-1} + \left(1 + \frac{2N}{S}\right)\gamma^2 L^2\right] + \frac{2S}{N}\left(\frac{\sigma^2}{K} + \frac{1}{3}\eta^2 K^2 L^2\right) \\
&\leq \left(1 - \frac{S}{2N}\right)\phi_i^{t-1} + \frac{2N}{S}\gamma^2 L^2 + \frac{2S}{N}\left(\frac{\sigma^2}{K} + \frac{1}{3}\eta^2 K^2 L^2\right) \\
&\leq \left(1 - \frac{S}{2N}\right)^t\phi_i^0 + \left(\frac{2N}{S}\gamma^2 L^2 + \frac{2S}{N}\left(\frac{\sigma^2}{K} + \frac{1}{3}\eta^2 K^2 L^2\right)\right)\sum_{\tau=0}^{t-1}\left(1 - \frac{S}{2N}\right)^\tau \\
&\leq \left(1 - \frac{S}{2N}\right)^t\phi_i^0 + 4\left(\frac{N^2}{S^2}\gamma^2 L^2 + \frac{\sigma^2}{K} + \frac{1}{3}\eta^2 K^2 L^2\right).
\end{aligned}
$$

Since $\boldsymbol{c}_i^{-1} = \frac{1}{K}\sum_{k=0}^{K-1}\nabla F\left(\boldsymbol{\theta}^0;\boldsymbol{\xi}_i^{-1,k}\right)$, we have

$$
\begin{aligned}
\phi_i^0 &= \left(1 - \frac{S}{N}\right)\mathbb{E}\left\|\nabla f_i\left(\boldsymbol{\theta}^0\right) - \boldsymbol{c}_i^{-1}\right\|^2 + \frac{S}{N}\mathbb{E}\left\|\nabla f_i\left(\boldsymbol{\theta}^0\right) - \frac{1}{K}\sum_k\nabla F\left(\boldsymbol{\theta}_i^{0,k};\boldsymbol{\xi}_i^{0,k}\right)\right\|^2 \\
&\leq \left(1 - \frac{S}{N}\right)\frac{\sigma^2}{K} + \frac{2S}{N}\left(L^2\mathbb{E}\left\|\boldsymbol{\theta}^0 - \boldsymbol{\theta}_i^{0,k}\right\|^2 + \frac{\sigma^2}{K}\right) \\
&\leq \left(1 + \frac{S}{N}\right)\frac{\sigma^2}{K} + \frac{2S}{3N}\eta^2 K^2 L^2 \\
&\leq \frac{2\sigma^2}{K} + \frac{2S}{3N}\eta^2 K^2 L^2.
\end{aligned}
$$

Then, we have

$$
\begin{aligned}
\phi_i^t &\leq \left(\frac{2\sigma^2}{K} + \frac{2S}{3N}\eta^2 K^2 L^2\right)\left(1 - \frac{S}{2N}\right)^t + 4\left(\frac{N^2}{S^2}\gamma^2 L^2 + \frac{\sigma^2}{K} + \frac{1}{3}\eta^2 K^2 L^2\right) \\
&\leq \left(\frac{2\sigma^2}{K} + \frac{2S}{3N}\eta^2 K^2 L^2\right)\left(1 - \frac{S}{4N}\right)^{2t} + 4\left(\frac{N^2}{S^2}\gamma^2 L^2 + \frac{\sigma^2}{K} + \frac{1}{3}\eta^2 K^2 L^2\right),
\end{aligned}
$$

where we use the relation $1 - \frac{S}{2N} \leq \left(1 - \frac{S}{4N}\right)^2$. $\qquad\square$

Define $\mathcal{E}^t := \nabla f\left(\boldsymbol{\theta}^t\right) - \boldsymbol{g}^t$ and $\boldsymbol{u}^t := \nabla f\left(\boldsymbol{\theta}^t\right) - \nabla f\left(\boldsymbol{\theta}^{t-1}\right)$. From the update rule of momentum $\boldsymbol{g}^t$, we have

$$
\begin{aligned}
\mathcal{E}^t &= (1-\beta)\left(\nabla f\left(\boldsymbol{\theta}^t\right) - \boldsymbol{g}^{t-1}\right) + \beta\underbrace{\left(\nabla f\left(\boldsymbol{\theta}^t\right) - \boldsymbol{c}^{t-1} - \frac{1}{SK}\sum_{i\in\mathcal{S}_t,k}\left(\nabla F\left(\boldsymbol{\theta}_i^{t,k};\boldsymbol{\xi}_i^{t,k}\right) - \boldsymbol{c}_i^{t-1}\right)\right)}_{:=\boldsymbol{v}^t} \\
&= (1-\beta)\mathcal{E}^{t-1} + (1-\beta)\boldsymbol{u}^t + \beta\boldsymbol{v}^t \\
&= (1-\beta)^t\mathcal{E}^0 + \sum_{\tau=1}^t\boldsymbol{u}^\tau(1-\beta)^{t+1-\tau} + \sum_{\tau=1}^t\beta\boldsymbol{v}^\tau(1-\beta)^{t-\tau}.
\end{aligned}
$$

Based on the triangle inequality of $\ell_2$ norm and the concavity of the square root $(\cdot)^{\frac{1}{2}}$, we have

$$
\mathbb{E}\left\|\mathcal{E}^t\right\| \leq (1-\beta)^t\mathbb{E}\left\|\mathcal{E}^0\right\| + \sum_{\tau=1}^t\mathbb{E}\left\|\boldsymbol{u}^\tau\right\|(1-\beta)^{t+1-\tau} + \left(\mathbb{E}\left\|\sum_{\tau=1}^t\beta\boldsymbol{v}^\tau(1-\beta)^{t-\tau}\right\|^2\right)^{\frac{1}{2}}. \quad \text{(A5)}
$$

Since $c_i^{-1} = \frac{1}{K} \sum_{k=0}^{K-1} \nabla F\left(\boldsymbol{\theta}^0; \boldsymbol{\xi}_i^{-1,k}\right)$ for any $i$, $c^{-1} = \frac{1}{N} \sum_i c_i^{-1}$, and $g^{-1} = c^{-1}$, we have

$$
\begin{aligned}
\mathbb{E}\left\|\mathcal{E}^0\right\| = \mathbb{E}\left\| \nabla f\left(\boldsymbol{\theta}^0\right) - \frac{1}{NK} \sum_{i,k} \nabla F\left(\boldsymbol{\theta}^0; \boldsymbol{\xi}_i^{-1,k}\right) + \frac{\beta}{SK} \sum_{i \in \mathcal{S}_0, k} \left(\nabla F\left(\boldsymbol{\theta}_i^0; \boldsymbol{\xi}_i^{-1,k}\right) - \nabla F\left(\boldsymbol{\theta}_i^{0,k}; \boldsymbol{\xi}_i^{0,k}\right)\right) \right\| \\
\leq \frac{\sigma}{\sqrt{NK}} + \mathbb{E}\left\| \frac{\beta}{SK} \sum_{i \in \mathcal{S}_0, k} \left(\nabla F\left(\boldsymbol{\theta}_i^0; \boldsymbol{\xi}_i^{-1,k}\right) \mp \nabla f_i\left(\boldsymbol{\theta}_i^0\right) \mp \nabla f_i\left(\boldsymbol{\theta}_i^{0,k}\right) - \nabla F\left(\boldsymbol{\theta}_i^{0,k}; \boldsymbol{\xi}_i^{0,k}\right)\right) \right\| \\
\leq \frac{\sigma}{\sqrt{NK}} + \frac{\beta\sigma}{\sqrt{SK}} + \frac{\beta}{SK}\mathbb{E}\left[ \sum_{i \in \mathcal{S}_0, k} \left\| \nabla f_i\left(\boldsymbol{\theta}^0\right) - \nabla f_i\left(\boldsymbol{\theta}_i^{0,k}\right) \right\| \right] + \frac{\beta\sigma}{\sqrt{SK}} \\
\leq \frac{\beta L}{NK} \sum_{i,k} \mathbb{E}\left\| \boldsymbol{\theta}_i^{0,k} - \boldsymbol{\theta}^0 \right\| + \frac{3\sigma}{\sqrt{SK}} \\
\leq \frac{1}{2} \eta\beta KL + \frac{3\sigma}{\sqrt{SK}}, \quad\quad\quad\quad\quad\quad\quad\quad\quad\quad\quad\quad\quad\quad\quad\quad\quad\quad\quad\quad\quad\quad\quad\quad\quad \text{(A6)}
\end{aligned}
$$

where the last inequality uses the results in Lemma 1.

Additionally, for any $t$, we have

$$
\left\|\boldsymbol{u}^t\right\| = \left\| \nabla f\left(\boldsymbol{\theta}^{t+1}\right) - \nabla f\left(\boldsymbol{\theta}^t\right) \right\| \leq L\left\| \boldsymbol{\theta}^{t+1} - \boldsymbol{\theta}^t \right\| \leq \gamma L \left\| \frac{1}{SK} \sum_{i \in \mathcal{S}_t, k} \frac{\boldsymbol{g}_i^{t,k}}{\|\boldsymbol{g}_i^{t,k}\|} \right\| \leq \gamma L. \quad \text{(A7)}
$$

To proceed, we handle the last term in (A5). First, we have

$$
\begin{aligned}
\mathbb{E}\left\| \sum_{\tau=1}^t \beta\boldsymbol{v}^\tau (1-\beta)^{t-\tau} \right\|^2 = \sum_{\tau=1}^t \beta^2 \mathbb{E}\|\boldsymbol{v}^\tau\|^2 (1-\beta)^{2(t-\tau)} \\
+ \sum_{1 \leq \tau_1, \tau_2 \leq t, \tau_1 \neq \tau_2} \mathbb{E}\left\langle \beta\boldsymbol{v}^{\tau_1}(1-\beta)^{t-\tau_1}, \beta\boldsymbol{v}^{\tau_2}(1-\beta)^{t-\tau_2} \right\rangle. \quad \text{(A8)}
\end{aligned}
$$

Let $\mathcal{F}^0 \neq \emptyset$ and $\mathcal{F}_i^{t,k} := \sigma(\{\boldsymbol{\theta}_i^{t,j}\}_{0 \leq j \leq k} \cup \mathcal{F}^t)$ and $\mathcal{F}^{t+1} := \sigma(\cup_i \mathcal{F}_i^{t,K})$ for all $t \geq 0$, where $\sigma()$ indicates the $\sigma$-algebra. Let $\mathbb{E}[\cdot|\mathcal{F}^t]$ represent the expectation conditioned on the filtration $\mathcal{F}^t$ with respect to the random variables $\{\boldsymbol{\xi}_i^{t,k}\}_{1 \leq i \leq N, 0 \leq k < K}$ in the $t$-th iteration. Let $\mathbb{E}_{\boldsymbol{\xi}_i^{t,k}}[\cdot]$ represent the expectation taking over the random sample $\boldsymbol{\xi}_i^{t,k}$. Similarly, let $\mathbb{E}_{\mathcal{S}_t}[\cdot]$ represent the expectation taking over the uniformly sampled client set $\mathcal{S}_t$. The set $\mathcal{S}_t$ is independent across different $t$. Then, for any $t$, we have

$$
\begin{aligned}
\mathbb{E}[\boldsymbol{v}^t|\mathcal{F}^t] &= \mathbb{E}_{\{\boldsymbol{\xi}_i^{t,k}\}_{\forall i,k}, \mathcal{S}_t}[\boldsymbol{v}^t] \\
&= \mathbb{E}_{\{\boldsymbol{\xi}_i^{t,k}\}_{\forall i,k}} \left[ \nabla f\left(\boldsymbol{\theta}^t\right) - \boldsymbol{c}^{t-1} - \frac{1}{NK} \sum_{i,k} \left( \nabla F\left(\boldsymbol{\theta}_i^{t,k}; \boldsymbol{\xi}_i^{t,k}\right) - \boldsymbol{c}_i^{t-1} \right) \right] \\
&= \nabla f\left(\boldsymbol{\theta}^t\right) - \frac{1}{NK} \sum_{i,k} \nabla f_i\left(\boldsymbol{\theta}_i^{t,k}\right),
\end{aligned}
$$

where the last equality is based on Assumption 3 and the fact that $\frac{1}{N}\sum_{i=1}^{N} c_i^t = c^t$ for any $t$. Then, for any $0 \le t_1 < t_2 \le T-1$, we have

$$\mathbb{E}\left\langle \boldsymbol{v}^{t_1}, \boldsymbol{v}^{t_2} \right\rangle$$

$$= \mathbb{E}\left\langle \boldsymbol{v}^{t_1}, \mathbb{E}\left[\boldsymbol{v}^{t_2}|\mathcal{F}^{t_2}\right] \right\rangle$$

$$= \mathbb{E}\left\langle \frac{1}{SK}\sum_{i \in \mathcal{S}_t, k}\left(\nabla f_i\left(\boldsymbol{\theta}^{t_1}\right) - \nabla f_i\left(\boldsymbol{\theta}_i^{t_1,k}\right)\right), \frac{1}{NK}\sum_{i,k}\left(\nabla f_i\left(\boldsymbol{\theta}^{t_2}\right) - \nabla f_i\left(\boldsymbol{\theta}_i^{t_2,k}\right)\right) \right\rangle$$

$$+ \mathbb{E}\left\langle \frac{1}{SK}\sum_{i \in \mathcal{S}_t, k}\left(\nabla f_i\left(\boldsymbol{\theta}_i^{t_1,k}\right) - \nabla F\left(\boldsymbol{\theta}_i^{t_1,k}; \boldsymbol{\xi}_i^{t_1,k}\right)\right), \frac{1}{NK}\sum_{i,k}\left(\nabla f_i\left(\boldsymbol{\theta}^{t_2}\right) - \nabla f_i\left(\boldsymbol{\theta}_i^{t_2,k}\right)\right) \right\rangle$$

$$+ \mathbb{E}\left\langle \mathbb{E}_{\mathcal{S}_t}\left[\frac{1}{S}\sum_{i \in \mathcal{S}_t} \boldsymbol{c}^{t-1} - \boldsymbol{c}_i^{t-1}\right], \frac{1}{NK}\sum_{i,k}\left(\nabla f_i\left(\boldsymbol{\theta}^{t_2}\right) - \nabla f_i\left(\boldsymbol{\theta}_i^{t_2,k}\right)\right) \right\rangle$$

$$\le \frac{L^2}{2NK}\sum_{i,k}\mathbb{E}\left\|\boldsymbol{\theta}_i^{t_1,k} - \boldsymbol{\theta}^{t_1}\right\|^2 + \frac{L^2}{2NK}\sum_{i,k}\mathbb{E}\left\|\boldsymbol{\theta}_i^{t_2,k} - \boldsymbol{\theta}^{t_2}\right\|^2 + \frac{\sigma}{\sqrt{SK}}\frac{L}{NK}\sum_{i,k}\left\|\boldsymbol{\theta}_i^{t_2,k} - \boldsymbol{\theta}^{t_2}\right\|$$

$$\le \frac{1}{3}\eta^2 K^2 L^2 + \frac{\eta\sqrt{K}L\sigma}{2\sqrt{S}}. \tag{A9}$$

Further, based on Lemma 2, we have

$$\mathbb{E}\left\|\boldsymbol{v}^t\right\|^2 \le \mathbb{E}\left\|\nabla f\left(\boldsymbol{\theta}^t\right) - \frac{1}{NK}\sum_{i,k}\nabla F\left(\boldsymbol{\theta}_i^{t,k}; \boldsymbol{\xi}_i^{t,k}\right)\right\|^2$$

$$+ \frac{1}{S}\underbrace{\frac{1}{N}\sum_i \mathbb{E}\left\|\frac{1}{K}\sum_k\left(\nabla F\left(\boldsymbol{\theta}_i^{t,k}; \boldsymbol{\xi}_i^{t,k}\right) - \frac{1}{N}\sum_i \nabla F\left(\boldsymbol{\theta}_i^{t,k}; \boldsymbol{\xi}_i^{t,k}\right)\right) - \left(\boldsymbol{c}_i^{t-1} - \boldsymbol{c}^{t-1}\right)\right\|^2}_{:=\Lambda_t}$$

$$\le 2\left(L^2\mathbb{E}\left\|\boldsymbol{\theta}^t - \boldsymbol{\theta}_i^{t,k}\right\|^2 + \frac{\sigma^2}{NK}\right) + \frac{\Lambda_t}{S}$$

$$\le \frac{2}{3}\eta^2 K^2 L^2 + \frac{2\sigma^2}{NK} + \frac{\Lambda_t}{S}.$$

$$\Lambda_t \le \frac{1}{N}\sum_i \mathbb{E}\left\|\frac{1}{K}\sum_k \nabla F\left(\boldsymbol{\theta}_i^{t,k}; \boldsymbol{\xi}_i^{t,k}\right) - \boldsymbol{c}_i^{t-1}\right\|^2$$

$$= \frac{1}{N}\sum_i \mathbb{E}\left\|\frac{1}{K}\sum_k \nabla F\left(\boldsymbol{\theta}_i^{t,k}; \boldsymbol{\xi}_i^{t,k}\right) \mp \nabla f_i\left(\boldsymbol{\theta}_i^{t,k}\right) \mp \nabla f_i\left(\boldsymbol{\theta}^t\right) \mp \nabla f_i\left(\boldsymbol{\theta}^{t-1}\right) - \boldsymbol{c}_i^{t-1}\right\|^2$$

$$\le \frac{4\sigma^2}{K} + \frac{4L^2}{NK}\sum_{i,k}\mathbb{E}\left\|\boldsymbol{\theta}_i^{t,k} - \boldsymbol{\theta}^t\right\|^2 + 4L^2\mathbb{E}\left\|\boldsymbol{\theta}^t - \boldsymbol{\theta}^{t-1}\right\|^2 + \frac{4}{N}\sum_i \mathbb{E}\left\|\nabla f_i\left(\boldsymbol{\theta}^{t-1}\right) - \boldsymbol{c}_i^{t-1}\right\|^2$$

$$\le \frac{4\sigma^2}{K} + \frac{4}{3}\eta^2 K^2 L^2 + 4\gamma^2 L^2 + \frac{4}{N}\sum_i \phi_i^{t-1},$$

where $\phi_i^{t-1} := \mathbb{E}\left\|\nabla f_i\left(\boldsymbol{\theta}^{t-1}\right) - \boldsymbol{c}_i^{t-1}\right\|^2$. From Lemma 5, we know that, for any $i$,

$$\phi_i^{t-1} \le \left(\frac{2\sigma^2}{K} + \frac{2S}{3N}\eta^2 K^2 L^2\right)\left(1 - \frac{S}{4N}\right)^{2(t-1)} + 4\left(\frac{N^2}{S^2}\gamma^2 L^2 + \frac{\sigma^2}{K} + \frac{1}{3}\eta^2 K^2 L^2\right)$$

$$\le \frac{6\sigma^2}{K} + 2\eta^2 K^2 L^2 + \frac{4N^2}{S^2}\gamma^2 L^2. \tag{A10}$$

Plugging the upper bound of $\phi_i^{t-1}$ into $\Lambda_t$ yields

$$\Lambda_t \le \frac{28\sigma^2}{K} + \frac{28}{3}\eta^2 K^2 L^2 + 4\gamma^2 L^2\left(1 + \frac{4N^2}{S^2}\right).$$

Then, we have

$$\mathbb{E}\left\|\boldsymbol{v}^t\right\|^2 \le \left(\frac{2}{3} + \frac{10}{S}\right)\eta^2 K^2 L^2 + \frac{30\sigma^2}{SK} + \frac{4\gamma^2 L^2}{S}\left(1 + \frac{4N^2}{S^2}\right). \tag{A11}$$

Plugging (A9) and (A11) into (A8) gives

$$\mathbb{E}\left\|\sum_{\tau=1}^{t}\beta\boldsymbol{v}^\tau(1-\beta)^{t-\tau}\right\|^2 \le \beta\mathbb{E}\|\boldsymbol{v}^\tau\|^2 + \mathbb{E}\left\langle\boldsymbol{v}^{\tau_1}, \boldsymbol{v}^{\tau_2}\right\rangle$$

$$\le \left(\frac{2}{3} + \frac{10}{S}\right)\beta\eta^2 K^2 L^2 + \frac{30\sigma^2\beta}{SK} + \frac{4\beta\gamma^2 L^2}{S}\left(1 + \frac{4N^2}{S^2}\right)$$

$$+ \frac{1}{3}\eta^2 K^2 L^2 + \frac{\eta\sqrt{K}L\sigma}{2\sqrt{S}}.$$

Since $\beta \le 1$, taking square root on both sides of the above inequality yields

$$\left(\mathbb{E}\left\|\sum_{\tau=1}^{t}\beta\boldsymbol{v}^\tau(1-\beta)^{t-\tau}\right\|^2\right)^{\frac{1}{2}}$$

$$\le \sqrt{1 + \frac{10\beta}{S}}\eta KL + \sigma\sqrt{\frac{30\beta}{SK}} + 2\gamma L\sqrt{\frac{\beta}{S}\left(1 + \frac{4N^2}{S^2}\right)} + \sqrt{\frac{\eta\sqrt{K}L\sigma}{2\sqrt{S}}}, \tag{A12}$$

where we use the fact that $\sqrt{a+b} \le \sqrt{a} + \sqrt{b}$, for any $a, b \ge 0$.

Plugging (A6), (A7), and (A12) into (A5), we have

$$\mathbb{E}\|\mathcal{E}^t\| \le (1-\beta)^t\left(\frac{1}{2}\eta\beta KL + \frac{3\sigma}{\sqrt{SK}}\right) + \frac{\gamma L}{\beta} + \sqrt{1 + \frac{10\beta}{S}}\eta KL + \sigma\sqrt{\frac{30\beta}{SK}}$$

$$+ 2\gamma L\sqrt{\frac{\beta}{S}\left(1 + \frac{4N^2}{S^2}\right)} + \sqrt{\frac{\eta\sqrt{K}L\sigma}{2\sqrt{S}}}.$$

Summing the above inequality over $t$ yields

$$\frac{1}{T}\sum_{t=0}^{T-1}\mathbb{E}\|\mathcal{E}^t\| \le \frac{1}{\beta T}\left(\frac{1}{2}\eta\beta KL + \frac{3\sigma}{\sqrt{SK}}\right) + \frac{\gamma L}{\beta} + \sqrt{1 + \frac{10\beta}{S}}\eta KL + \sigma\sqrt{\frac{30\beta}{SK}}$$

$$+ 2\gamma L\sqrt{\frac{\beta}{S}\left(1 + \frac{4N^2}{S^2}\right)} + \sqrt{\frac{\eta\sqrt{K}L\sigma}{2\sqrt{S}}}.$$

### A.2 PROOF OF LEMMA 4

Recall that $\boldsymbol{g}_i^{t,k} = \beta\left(\nabla F\left(\boldsymbol{\theta}_i^{t,k}; \boldsymbol{\xi}_i^{t,k}\right) - \boldsymbol{c}_i^{t-1} + \boldsymbol{c}^{t-1}\right) + (1-\beta)\boldsymbol{g}^{t-1}$, and

$$\boldsymbol{g}^t = \frac{1}{SK}\sum_{i\in\mathcal{S}_t,k}\boldsymbol{g}_i^{t,k}$$

$$= \beta\left(\frac{1}{S}\sum_{i\in\mathcal{S}_t}\left(\frac{1}{K}\sum_{k=0}^{K-1}\nabla F\left(\boldsymbol{\theta}_i^{t,k}; \boldsymbol{\xi}_i^{t,k}\right) - \boldsymbol{c}_i^{t-1}\right) + \boldsymbol{c}^{t-1}\right) + (1-\beta)\boldsymbol{g}^{t-1}.$$

Then, we have

$$
\mathbb{E}\left[\frac{1}{SK}\sum_{i\in\mathcal{S}_t,k}\left\|\boldsymbol{g}_i^{t,k}-\boldsymbol{g}^t\right\|\right]
$$

$$
=\beta\mathbb{E}\left[\frac{1}{SK}\sum_{i\in\mathcal{S}_t,k}\left\|\nabla F\left(\boldsymbol{\theta}_i^{t,k};\boldsymbol{\xi}_i^{t,k}\right)-\boldsymbol{c}_i^{t-1}-\frac{1}{S}\sum_{i\in\mathcal{S}_t}\left(\frac{1}{K}\sum_{k=0}^{K-1}\nabla F\left(\boldsymbol{\theta}_i^{t,k};\boldsymbol{\xi}_i^{t,k}\right)-\boldsymbol{c}_i^{t-1}\right)\right\|\right]
$$

$$
\leq\frac{2\beta}{NK}\sum_{i,k}\mathbb{E}\left\|\nabla F\left(\boldsymbol{\theta}_i^{t,k};\boldsymbol{\xi}_i^{t,k}\right)-\boldsymbol{c}_i^{t-1}\right\|
$$

$$
=\frac{2\beta}{NK}\sum_{i,k}\mathbb{E}\left\|\nabla F\left(\boldsymbol{\theta}_i^{t,k};\boldsymbol{\xi}_i^{t,k}\right)\mp\nabla f_i\left(\boldsymbol{\theta}_i^{t,k}\right)\mp\nabla f_i\left(\boldsymbol{\theta}^t\right)\mp\nabla f_i\left(\boldsymbol{\theta}^{t-1}\right)-\boldsymbol{c}_i^{t-1}\right\|
$$

$$
\leq 2\beta\left(\sigma+\frac{L}{NK}\sum_{i,k}\left\|\boldsymbol{\theta}_i^{t,k}-\boldsymbol{\theta}^t\right\|+L\left\|\boldsymbol{\theta}^t-\boldsymbol{\theta}^{t-1}\right\|\right)+\frac{2\beta}{N}\sum_i\mathbb{E}\left\|\nabla f_i\left(\boldsymbol{\theta}^{t-1}\right)-\boldsymbol{c}_i^{t-1}\right\|
$$

$$
\leq 2\beta\left(\sigma+\frac{1}{2}\eta KL+\gamma L\right)+\frac{2\beta}{N}\sum_i\sqrt{\phi_i^{t-1}}.
$$

From Lemma 5, we know that

$$
\sqrt{\phi_i^{t-1}}\leq\left(\frac{\sqrt{2}\sigma}{\sqrt{K}}+\sqrt{\frac{2S}{3N}}\eta KL\right)\left(1-\frac{S}{4N}\right)^{t-1}+2\left(\frac{N}{S}\gamma L+\frac{\sigma}{\sqrt{K}}+\frac{1}{\sqrt{3}}\eta KL\right),\forall i.
$$

Thus, we have

$$
\mathbb{E}\left[\frac{1}{SK}\sum_{i\in\mathcal{S}_t,k}\left\|\boldsymbol{g}_i^{t,k}-\boldsymbol{g}^t\right\|\right]\leq 2\beta\left(\left(1+\frac{2}{\sqrt{K}}\right)\sigma+2\eta KL+\left(1+\frac{2N}{S}\right)\gamma L\right)
$$

$$
+2\beta\left(\frac{\sqrt{2}\sigma}{\sqrt{K}}+\sqrt{\frac{2S}{3N}}\eta KL\right)\left(1-\frac{S}{4N}\right)^{t-1}.
$$

Summing the above inequality over $t$ yields

$$
\frac{1}{SKT}\sum_{t=1}^{T}\mathbb{E}\left[\sum_{i\in\mathcal{S}_t,k}\left\|\boldsymbol{g}_i^{t,k}-\boldsymbol{g}^t\right\|\right]\leq 2\beta\left(\left(1+\frac{2}{\sqrt{K}}\right)\sigma+2\eta KL+\left(1+\frac{2N}{S}\right)\gamma L\right)
$$

$$
+\frac{8N\beta}{ST}\left(\frac{\sqrt{2}\sigma}{\sqrt{K}}+\sqrt{\frac{2S}{3N}}\eta KL\right).
$$

# B   THEORETICAL ANALYSIS OF PADAMFED WITH VARIANCE REDUCTION

The analysis of PAdaMFed-VR is similar to that of PAdaMFed. We first present the following two auxiliary Lemmas.

**Lemma 6.** *Under Assumptions 1 and 3, the disparity $\frac{1}{T}\sum_{t=0}^{T-1}\mathbb{E}\|\nabla f\left(\boldsymbol{\theta}^t\right)-\boldsymbol{g}^t\|$ is upper bounded by:*

$$
\frac{1}{T}\sum_{t=0}^{T-1}\mathbb{E}\left\|\nabla f\left(\boldsymbol{\theta}^t\right)-\boldsymbol{g}^t\right\|\leq\frac{1}{\beta T}\left(\frac{1}{2}\eta KL+\frac{3\sigma}{\sqrt{SK}}\right)+\frac{\eta KL}{2\beta}+\gamma L\sqrt{\frac{2}{SK\beta}}+\sigma\sqrt{\frac{22\beta}{SK}}
$$

$$
+\gamma L\sqrt{\frac{3\beta}{S}\left(1+\frac{4N^2}{S^2}\right)}+\eta KL\sqrt{\frac{6\beta}{S}}.
$$

---

**Algorithm 2** PAdaMFed-VR: PAdaMFed with Variance Reduction

---

1: **Require:** initial model $\boldsymbol{\theta}^0$, $\boldsymbol{\theta}^{-1} = \boldsymbol{\theta}^0$, control variates $\boldsymbol{c}_i^{-1} = \frac{1}{K} \sum_{k=0}^{K-1} \nabla F\left(\boldsymbol{\theta}^0; \boldsymbol{\xi}_i^{-1,k}\right)$ for any $i$, $\boldsymbol{c}^{-1} = \frac{1}{N} \sum_i \boldsymbol{c}_i^{-1}$, momentum $\boldsymbol{g}^{-1} = \boldsymbol{c}^{-1}$, global learning rate $\gamma$, local learning rate $\eta$, and momentum parameter $\beta$
2: **for** $t = 0, \cdots, T - 1$ **do**
3:    **Central Server:** Uniformly sample clients $\mathcal{S}_t \subseteq \{1, \cdots, N\}$ with $|\mathcal{S}_t| = S$
4:    **for** each client $i \in \mathcal{S}_t$ in parallel **do**
5:       Initialize local model $\boldsymbol{\theta}_i^{t,0} = \boldsymbol{\theta}^t$ and control variate $\boldsymbol{c}_i^t = \boldsymbol{0}$ (for $i \notin \mathcal{S}_t$, $\boldsymbol{c}_i^t = \boldsymbol{c}_i^{t-1}$ )
6:       **for** $k = 0, \cdots, K - 1$ **do**
7:          Compute $\boldsymbol{g}_i^{t,k} = \nabla F\left(\boldsymbol{\theta}_i^{t,k}; \boldsymbol{\xi}_i^{t,k}\right) + \beta\left(\boldsymbol{c}^{t-1} - \boldsymbol{c}_i^{t-1}\right) + (1 - \beta)\left(\boldsymbol{g}^{t-1} - \nabla F\left(\boldsymbol{\theta}^{t-1}; \boldsymbol{\xi}_i^{t,k}\right)\right)$
8:          Update local model $\boldsymbol{\theta}_i^{t,k+1} = \boldsymbol{\theta}_i^{t,k} - \eta \frac{\boldsymbol{g}_i^{t,k}}{\|\boldsymbol{g}_i^{t,k}\|}$
9:          Update control variate $\boldsymbol{c}_i^t = \boldsymbol{c}_i^t + \frac{1}{K} \nabla F\left(\boldsymbol{\theta}_i^{t,k}; \boldsymbol{\xi}_i^{t,k}\right)$
10:       **end for**
11:       Upload $\boldsymbol{\theta}_i^{t,K}$ and $\boldsymbol{c}_i^t$ to central server
12:    **end for**
     **Central server:**
13:    Aggregate local updates $\overline{\boldsymbol{g}}^t = \frac{1}{\eta SK} \sum_{i \in \mathcal{S}_t} \left(\boldsymbol{\theta}^t - \boldsymbol{\theta}_i^{t,K}\right)$
14:    Update global model $\boldsymbol{\theta}^{t+1} = \boldsymbol{\theta}^t - \gamma \overline{\boldsymbol{g}}^t$
15:    Aggregate control variate $\boldsymbol{c}^t = \boldsymbol{c}^{t-1} + \frac{1}{N} \sum_{i \in \mathcal{S}_t} \left(\boldsymbol{c}_i^t - \boldsymbol{c}_i^{t-1}\right)$
16:    Aggregate momentum $\boldsymbol{g}^t = \beta\left(\frac{1}{S} \sum_{i \in \mathcal{S}_t} \left(\boldsymbol{c}_i^t - \boldsymbol{c}_i^{t-1}\right) + \boldsymbol{c}^{t-1}\right) + (1 - \beta)\boldsymbol{g}^{t-1}$
17:    Download $\boldsymbol{\theta}^{t+1}$, $\beta \boldsymbol{c}^t + (1 - \beta)\boldsymbol{g}^t$ to all clients
18: **end for**

---

**Lemma 7.** *Under Assumptions 1 and 3, the gradient dissimilarity* $\frac{1}{SKT} \sum_{t=0}^{T-1} \mathbb{E}\left[\sum_{i \in \mathcal{S}_t} \left\|\boldsymbol{g}_i^{t,k} - \boldsymbol{g}^t\right\|\right]$ *is upper bounded by:*

$$\frac{1}{SKT} \sum_{t=1}^{T} \mathbb{E}\left[\sum_{i \in \mathcal{S}_t, k} \left\|\boldsymbol{g}_i^{t,k} - \boldsymbol{g}^t\right\|\right] \leq 2\beta\left(\left(1 + \frac{2}{\sqrt{K}}\right)\sigma + 2\eta KL + \left(1 + \frac{2N}{S}\right)\gamma L\right)$$
$$+ \frac{8N\beta}{ST}\left(\frac{\sqrt{2}\sigma}{\sqrt{K}} + \sqrt{\frac{2S}{3N}}\eta KL\right) + \eta KL + 2\gamma L.$$

Set $\beta = \frac{\beta_0}{T^{\frac{2}{3}}}$ and $\gamma = \frac{\gamma_0}{T^{\frac{2}{3}}}$. $\eta = \frac{1}{KT}$. From Lemma 6, we know that

$$\frac{1}{T} \sum_{t=0}^{T-1} \mathbb{E}\left\|\nabla f\left(\boldsymbol{\theta}^t\right) - \boldsymbol{g}^t\right\| \leq \frac{1}{\beta_0 T^{\frac{1}{3}}}\left(\frac{L}{2T} + \frac{3\sigma}{\sqrt{SK}}\right) + \frac{L}{2\beta_0 T^{\frac{1}{3}}} + \frac{\gamma_0 L}{T^{\frac{1}{3}}}\sqrt{\frac{2}{SK\beta_0}} + \frac{\sigma}{T^{\frac{1}{3}}}\sqrt{\frac{22\beta_0}{SK}}$$
$$+ \frac{\gamma_0 L}{T}\sqrt{\frac{3\beta_0}{S}\left(1 + \frac{4N^2}{S^2}\right)} + \frac{L}{T^{\frac{4}{3}}}\sqrt{\frac{6\beta_0}{S}}$$
$$\lesssim \frac{3\sigma}{\beta_0 \sqrt{SK}T^{\frac{1}{3}}} + \frac{L}{2\beta_0 T^{\frac{1}{3}}} + \frac{\gamma_0 L}{T^{\frac{1}{3}}}\sqrt{\frac{2}{SK\beta_0}} + \frac{\sigma}{T^{\frac{1}{3}}}\sqrt{\frac{22\beta_0}{SK}}. \quad (A13)$$

Similarly, from Lemma 7, we have

$$\frac{1}{SKT} \sum_{t=1}^{T} \mathbb{E}\left[\sum_{i \in \mathcal{S}_t, k} \left\|\boldsymbol{g}_i^{t,k} - \boldsymbol{g}^t\right\|\right] \leq \frac{2\beta_0}{T^{\frac{2}{3}}}\left(\left(1 + \frac{2}{\sqrt{K}}\right)\sigma + \frac{2L}{T} + \left(1 + \frac{2N}{S}\right)\frac{\gamma_0 L}{T^{\frac{2}{3}}}\right)$$
$$+ \frac{8N\beta_0}{ST^{\frac{5}{3}}}\left(\frac{\sqrt{2}\sigma}{\sqrt{K}} + \sqrt{\frac{2S}{3N}}\frac{L}{T}\right) + \frac{L}{T} + \frac{2\gamma_0 L}{T^{\frac{2}{3}}}. \quad (A14)$$

Plugging (A13) and (A14) into (A2), we have

$$\frac{1}{T}\sum_{t=0}^{T-1}\mathbb{E}\left\|\nabla f\left(\boldsymbol{\theta}^t\right)\right\| \lesssim \frac{\Delta}{\gamma_0 T^{\frac{1}{3}}} + \frac{6\sigma}{\beta_0\sqrt{SK}T^{\frac{1}{3}}} + \frac{L}{\beta_0 T^{\frac{1}{3}}} + \frac{2\sqrt{2}\gamma_0 L}{\sqrt{SK}\beta_0 T^{\frac{1}{3}}} + \frac{2\sigma\sqrt{22\beta_0}}{\sqrt{SK}T^{\frac{1}{3}}}$$
$$+ \frac{2\beta_0\sigma}{T^{\frac{2}{3}}} + \frac{2\gamma_0 L}{T^{\frac{2}{3}}}.$$

Set $\beta_0 = (SK)^{\frac{1}{3}}$ and $\gamma_0 = (SK)^{\frac{1}{3}}$, we have

$$\frac{1}{T}\sum_{t=0}^{T-1}\mathbb{E}\left\|\nabla f\left(\boldsymbol{\theta}^t\right)\right\| \lesssim \frac{\Delta}{\gamma_0 T^{\frac{1}{3}}} + \frac{6\sigma}{\beta_0\sqrt{SK}T^{\frac{1}{3}}} + \frac{L}{\beta_0 T^{\frac{1}{3}}} + \frac{2\sqrt{2}\gamma_0 L}{\sqrt{SK}\beta_0 T^{\frac{1}{3}}}$$
$$+ \frac{2\sigma\sqrt{22\beta_0}}{\sqrt{SK}T^{\frac{1}{3}}} + \frac{2\beta_0\sigma}{T^{\frac{2}{3}}} + \frac{2\gamma_0 L}{T^{\frac{2}{3}}}$$
$$\leq \mathcal{O}\left(\frac{\Delta + L + \sigma}{(SKT)^{\frac{1}{3}}} + \frac{(L+\sigma)(SK)^{\frac{1}{3}}}{T^{\frac{2}{3}}}\right).$$

By setting $SK \leq \mathcal{O}\left(\sqrt{T}\right)$, we have $\frac{(SK)^{\frac{1}{3}}}{T^{\frac{2}{3}}} \propto \mathcal{O}\left((SKT)^{-\frac{1}{3}}\right)$ and thus

$$\frac{1}{T}\sum_{t=0}^{T-1}\mathbb{E}\left\|\nabla f\left(\boldsymbol{\theta}^t\right)\right\| \leq \mathcal{O}\left(\frac{\Delta + L + \sigma}{(SKT)^{\frac{1}{3}}}\right).$$

### B.1 PROOF OF LEMMA 6

Since $\mathcal{E}^t := \nabla f\left(\boldsymbol{\theta}^t\right) - \boldsymbol{g}^t$, we have

$$\mathcal{E}^t = \nabla f\left(\boldsymbol{\theta}^t\right) - \frac{1}{SK}\sum_{i\in\mathcal{S}_t,k}\nabla F\left(\boldsymbol{\theta}_i^{t,k};\boldsymbol{\xi}_i^{t,k}\right) + \frac{\beta}{S}\sum_{i\in\mathcal{S}_t}\left(\boldsymbol{c}_i^{t-1} - \boldsymbol{c}^{t-1}\right)$$

$$- (1-\beta)\left(\boldsymbol{g}^{t-1} \mp \nabla f\left(\boldsymbol{\theta}^{t-1}\right) - \frac{1}{SK}\sum_{i\in\mathcal{S}_t,k}\nabla F\left(\boldsymbol{\theta}^{t-1};\boldsymbol{\xi}_i^{t,k}\right)\right)$$

$$= (1-\beta)\mathcal{E}^{t-1} + \underbrace{\frac{1}{SK}\sum_{i\in\mathcal{S}_t,k}\left(\nabla F\left(\boldsymbol{\theta}^t;\boldsymbol{\xi}_i^{t,k}\right) - \nabla F\left(\boldsymbol{\theta}_i^{t,k};\boldsymbol{\xi}_i^{t,k}\right)\right)}_{:=\boldsymbol{w}^t}$$

$$+ \beta\underbrace{\left(\nabla f\left(\boldsymbol{\theta}^t\right) - \boldsymbol{c}^{t-1} - \frac{1}{SK}\sum_{i\in\mathcal{S}_t,k}\left(\nabla F\left(\boldsymbol{\theta}^t;\boldsymbol{\xi}_i^{t,k}\right) - \boldsymbol{c}_i^{t-1}\right)\right)}_{:=\widetilde{\boldsymbol{v}}^t}$$

$$+ (1-\beta)\underbrace{\left(\frac{1}{SK}\sum_{i\in\mathcal{S}_t,k}\left(\nabla F\left(\boldsymbol{\theta}^{t-1};\boldsymbol{\xi}_i^{t,k}\right) - \nabla F\left(\boldsymbol{\theta}^t;\boldsymbol{\xi}_i^{t,k}\right)\right) + \nabla f\left(\boldsymbol{\theta}^t\right) - \nabla f\left(\boldsymbol{\theta}^{t-1}\right)\right)}_{:=\widetilde{\boldsymbol{u}}^t}$$

$$= (1-\beta)^t\mathcal{E}^0 + \sum_{\tau=1}^t\boldsymbol{w}^\tau(1-\beta)^{t-\tau} + \sum_{\tau=1}^t\widetilde{\boldsymbol{u}}^\tau(1-\beta)^{t+1-\tau} + \sum_{\tau=1}^t\beta\widetilde{\boldsymbol{v}}^\tau(1-\beta)^{t-\tau}.$$

$$\mathbb{E}\left\|\mathcal{E}^t\right\| \leq (1-\beta)^t\mathbb{E}\left\|\mathcal{E}^0\right\| + \sum_{\tau=1}^t\mathbb{E}\left\|\boldsymbol{w}^t\right\|(1-\beta)^{t-\tau} + \left(\mathbb{E}\left\|\sum_{\tau=1}^t\widetilde{\boldsymbol{u}}^\tau(1-\beta)^{t+1-\tau}\right\|^2\right)^{\frac{1}{2}}$$

$$+ \left(\mathbb{E}\left\|\sum_{\tau=1}^t\beta\widetilde{\boldsymbol{v}}^\tau(1-\beta)^{t-\tau}\right\|^2\right)^{\frac{1}{2}}. \tag{A15}$$

Since $\boldsymbol{\theta}^{-1} = \boldsymbol{\theta}^0$, $\boldsymbol{c}_i^{-1} = \frac{1}{K} \sum_{k=0}^{K-1} \nabla F\left(\boldsymbol{\theta}^0; \boldsymbol{\xi}_i^{-1,k}\right)$ for any $i$, $\boldsymbol{c}^{-1} = \frac{1}{N} \sum_i \boldsymbol{c}_i^{-1}$, and $\boldsymbol{g}^{-1} = \boldsymbol{c}^{-1}$, we have

$$
\begin{aligned}
\mathbb{E}\left\|\mathcal{E}^0\right\| =& \mathbb{E}\left\| \nabla f\left(\boldsymbol{\theta}^0\right) - \frac{1}{NK} \sum_{i,k} \nabla F\left(\boldsymbol{\theta}^0; \boldsymbol{\xi}_i^{-1,k}\right) \right. \\
& \left. + \frac{1}{SK} \sum_{i \in \mathcal{S}_0, k} \left( \beta \nabla F\left(\boldsymbol{\theta}_i^0; \boldsymbol{\xi}_i^{-1,k}\right) + (1-\beta) \nabla F\left(\boldsymbol{\theta}^0; \boldsymbol{\xi}_i^{0,k}\right) - \nabla F\left(\boldsymbol{\theta}_i^{0,k}; \boldsymbol{\xi}_i^{0,k}\right) \right) \right\| \\
\leq& \frac{\sigma}{\sqrt{NK}} + \frac{\sigma}{\sqrt{SK}} + \frac{1}{SK} \mathbb{E}\left[ \sum_{i \in \mathcal{S}_0, k} \left\| \nabla f_i\left(\boldsymbol{\theta}^0\right) - \nabla f_i\left(\boldsymbol{\theta}_i^{0,k}\right) \right\| \right] + \frac{\sigma}{\sqrt{SK}} \\
\leq& \frac{L}{NK} \sum_{i,k} \mathbb{E}\left\| \boldsymbol{\theta}_i^{0,k} - \boldsymbol{\theta}^0 \right\| + \frac{3\sigma}{\sqrt{SK}} \\
\leq& \frac{1}{2} \eta K L + \frac{3\sigma}{\sqrt{SK}}.
\end{aligned}
\tag{A16}
$$

Then, we have

$$
\left\| \boldsymbol{w}^t \right\| \leq \frac{L}{SK} \sum_{i \in \mathcal{S}_t, k} \mathbb{E}\left\| \boldsymbol{\theta}_i^{t,k} - \boldsymbol{\theta}^t \right\| \leq \frac{1}{2} \eta K L.
\tag{A17}
$$

Additionally, since $\mathbb{E}[\widetilde{\boldsymbol{u}}^t | \mathcal{F}^t] = \boldsymbol{0}$, then, for any $0 \leq t_1 < t_2 \leq T-1$, we have

$$
\mathbb{E}\left\langle \widetilde{\boldsymbol{u}}^{t_1}, \widetilde{\boldsymbol{u}}^{t_2} \right\rangle = \mathbb{E}\left\langle \widetilde{\boldsymbol{u}}^{t_1}, \mathbb{E}[\widetilde{\boldsymbol{u}}^{t_2} | \mathcal{F}^{t_2}] \right\rangle = 0.
$$

From Lemma 2, for any $t$, we have

$$
\begin{aligned}
\mathbb{E}\left\|\widetilde{\boldsymbol{u}}^t\right\|^2 \leq& \mathbb{E}\left\| \frac{1}{NK} \sum_{i,k} \left( \nabla F\left(\boldsymbol{\theta}^{t-1}; \boldsymbol{\xi}_i^{t,k}\right) - \nabla F\left(\boldsymbol{\theta}^t; \boldsymbol{\xi}_i^{t,k}\right) \right) + \nabla f\left(\boldsymbol{\theta}^t\right) - \nabla f\left(\boldsymbol{\theta}^{t-1}\right) \right\|^2 \\
& + \frac{1}{SN} \sum_{i=1}^N \mathbb{E}\left\| \frac{1}{K} \sum_k \left( \nabla F\left(\boldsymbol{\theta}^{t-1}; \boldsymbol{\xi}_i^{t,k}\right) - \nabla F\left(\boldsymbol{\theta}^t; \boldsymbol{\xi}_i^{t,k}\right) \right) + \nabla f\left(\boldsymbol{\theta}^t\right) - \nabla f\left(\boldsymbol{\theta}^{t-1}\right) \right\|^2 \\
\leq& \frac{L^2}{NK} \left\| \boldsymbol{\theta}^t - \boldsymbol{\theta}^{t-1} \right\|^2 + \frac{L^2}{SK} \left\| \boldsymbol{\theta}^t - \boldsymbol{\theta}^{t-1} \right\|^2 \\
\leq& \frac{2\gamma^2 L^2}{SK}.
\end{aligned}
$$

Then, we have

$$
\mathbb{E}\left\| \sum_{\tau=1}^t \widetilde{\boldsymbol{u}}^\tau (1-\beta)^{t+1-\tau} \right\|^2 = \sum_{\tau=1}^t \mathbb{E}\|\widetilde{\boldsymbol{u}}^\tau\|^2 (1-\beta)^{t+1-\tau} \leq \frac{2\gamma^2 L^2}{SK\beta}.
\tag{A18}
$$

Similarly, since $\mathbb{E}[\widetilde{\boldsymbol{v}}^t | \mathcal{F}^t] = \boldsymbol{0}$, for any $0 \leq t_1 < t_2 \leq T-1$, we have

$$
\mathbb{E}\left\langle \widetilde{\boldsymbol{v}}^{t_1}, \widetilde{\boldsymbol{v}}^{t_2} \right\rangle = \mathbb{E}\left\langle \widetilde{\boldsymbol{v}}^{t_1}, \mathbb{E}[\widetilde{\boldsymbol{v}}^{t_2} | \mathcal{F}^{t_2}] \right\rangle = 0.
$$

From Lemma 2, for any $t$, we have

$$
\begin{aligned}
\mathbb{E}\|\widetilde{\boldsymbol{v}}^t\|^2 \leq& \mathbb{E}\left\| \nabla f\left(\boldsymbol{\theta}^t\right) - \frac{1}{NK} \sum_{i,k} \nabla F\left(\boldsymbol{\theta}^t; \boldsymbol{\xi}_i^{t,k}\right) \right\|^2 + \frac{1}{SN} \sum_{i=1}^N \left\| \frac{1}{K} \sum_k \nabla F\left(\boldsymbol{\theta}^t; \boldsymbol{\xi}_i^{t,k}\right) - \boldsymbol{c}_i^{t-1} \right\|^2 \\
\leq& \frac{\sigma^2}{NK} + \frac{1}{SN} \sum_{i=1}^N \left\| \frac{1}{K} \sum_k \nabla F\left(\boldsymbol{\theta}^t; \boldsymbol{\xi}_i^{t,k}\right) \mp \nabla f_i\left(\boldsymbol{\theta}^t\right) \mp \nabla f_i\left(\boldsymbol{\theta}^{t-1}\right) - \boldsymbol{c}_i^{t-1} \right\|^2 \\
\leq& \frac{\sigma^2}{NK} + \frac{3\sigma^2}{SK} + \frac{3}{S} \gamma^2 L^2 + \frac{3}{S} \frac{1}{N} \sum_{i=1}^N \phi_i^{t-1}.
\end{aligned}
$$

By (A10), $\phi_i^{t-1} \le \frac{6\sigma^2}{K} + 2\eta^2 K^2 L^2 + \frac{4N^2}{S^2}\gamma^2 L^2$. Then, we have

$$\mathbb{E}\|\widetilde{\boldsymbol{v}}^t\|^2 \le \frac{22\sigma^2}{SK} + \frac{3\gamma^2 L^2}{S}\left(1 + \frac{4N^2}{S^2}\right) + \frac{6}{S}\eta^2 K^2 L^2.$$

$$\mathbb{E}\left\|\sum_{\tau=1}^{t}\beta\widetilde{\boldsymbol{v}}^\tau(1-\beta)^{t-\tau}\right\|^2 \le \beta^2 \sum_{\tau=1}^{t}\mathbb{E}\|\widetilde{\boldsymbol{v}}^\tau\|^2(1-\beta)^{t+1-\tau}$$
$$\le \frac{22\beta\sigma^2}{SK} + \frac{3\beta\gamma^2 L^2}{S}\left(1 + \frac{4N^2}{S^2}\right) + \frac{6\beta}{S}\eta^2 K^2 L^2. \qquad (A19)$$

Plugging (A16), (A17), (A18), and (A19) into (A15) yields

$$\mathbb{E}\|\mathcal{E}^t\| \le (1-\beta)^t\left(\frac{1}{2}\eta KL + \frac{3\sigma}{\sqrt{SK}}\right) + \frac{\eta KL}{2\beta} + \gamma L\sqrt{\frac{2}{SK\beta}} + \sigma\sqrt{\frac{22\beta}{SK}}$$
$$+ \gamma L\sqrt{\frac{3\beta}{S}\left(1 + \frac{4N^2}{S^2}\right)} + \eta KL\sqrt{\frac{6\beta}{S}}.$$

Summing the above inequality over $t$ yields

$$\frac{1}{T}\sum_{t=0}^{T-1}\mathbb{E}\|\mathcal{E}^t\| \le \frac{1}{\beta T}\left(\frac{1}{2}\eta KL + \frac{3\sigma}{\sqrt{SK}}\right) + \frac{\eta KL}{2\beta} + \gamma L\sqrt{\frac{2}{SK\beta}} + \sigma\sqrt{\frac{22\beta}{SK}}$$
$$+ \gamma L\sqrt{\frac{3\beta}{S}\left(1 + \frac{4N^2}{S^2}\right)} + \eta KL\sqrt{\frac{6\beta}{S}}.$$

## B.2 Proof of Lemma 7

With variance reduction, we have $\boldsymbol{g}_i^{t,k} = \nabla F\left(\boldsymbol{\theta}_i^{t,k}; \boldsymbol{\xi}_i^{t,k}\right) - \beta\left(\boldsymbol{c}_i^{t-1} - \boldsymbol{c}^{t-1}\right) + (1-\beta)\left(\boldsymbol{g}^{t-1} - \nabla F\left(\boldsymbol{\theta}^{t-1}; \boldsymbol{\xi}_i^{t,k}\right)\right)$. Since $\boldsymbol{g}^t = \frac{1}{SK}\sum_{i\in\mathcal{S}_t,k}\boldsymbol{g}_i^{t,k}$, we have

$$\mathbb{E}\left[\frac{1}{SK}\sum_{i\in\mathcal{S}_t}\left\|\boldsymbol{g}_i^{t,k} - \boldsymbol{g}^t\right\|\right]$$
$$\le \frac{2}{NK}\sum_{i,k}\mathbb{E}\left\|\nabla F\left(\boldsymbol{\theta}_i^{t,k}; \boldsymbol{\xi}_i^{t,k}\right) - \beta\boldsymbol{c}_i^{t-1} - (1-\beta)\nabla F\left(\boldsymbol{\theta}^{t-1}; \boldsymbol{\xi}_i^{t,k}\right)\right\|$$
$$\le \frac{2}{NK}\sum_{i,k}\left(\beta\mathbb{E}\left\|\nabla F\left(\boldsymbol{\theta}_i^{t,k}; \boldsymbol{\xi}_i^{t,k}\right) - \boldsymbol{c}_i^{t-1}\right\| + (1-\beta)\mathbb{E}\left\|\nabla F\left(\boldsymbol{\theta}_i^{t,k}; \boldsymbol{\xi}_i^{t,k}\right) - \nabla F\left(\boldsymbol{\theta}^{t-1}; \boldsymbol{\xi}_i^{t,k}\right)\right\|\right)$$
$$\le \frac{2\beta}{NK}\sum_{i,k}\mathbb{E}\left\|\nabla F\left(\boldsymbol{\theta}_i^{t,k}; \boldsymbol{\xi}_i^{t,k}\right) - \boldsymbol{c}_i^{t-1}\right\| + 2L(1-\beta)\left(\frac{1}{NK}\sum_{i,k}\mathbb{E}\left\|\boldsymbol{\theta}_i^{t,k} - \boldsymbol{\theta}^t\right\| + \mathbb{E}\left\|\boldsymbol{\theta}^t - \boldsymbol{\theta}^{t-1}\right\|\right)$$
$$\le \frac{2\beta}{NK}\sum_{i,k}\mathbb{E}\left\|\nabla F\left(\boldsymbol{\theta}_i^{t,k}; \boldsymbol{\xi}_i^{t,k}\right) - \boldsymbol{c}_i^{t-1}\right\| + \eta KL + 2\gamma L.$$

From Section A.2, we know that

$$\frac{2\beta}{NK}\sum_{i,k}\mathbb{E}\left\|\nabla F\left(\boldsymbol{\theta}_i^{t,k}; \boldsymbol{\xi}_i^{t,k}\right) - \boldsymbol{c}_i^{t-1}\right\| \le 2\beta\left(\left(1 + \frac{2}{\sqrt{K}}\right)\sigma + 2\eta KL + \left(1 + \frac{2N}{S}\right)\gamma L\right)$$
$$+ 2\beta\left(\frac{\sqrt{2}\sigma}{\sqrt{K}} + \sqrt{\frac{2S}{3N}}\eta KL\right)\left(1 - \frac{S}{4N}\right)^{t-1}.$$

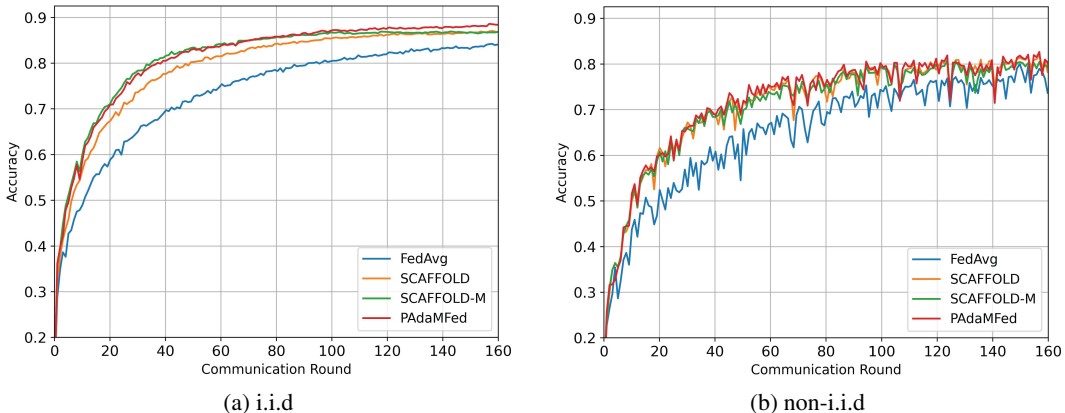

(a) i.i.d           (b) non-i.i.d

Figure 3: Test accuracy versus the number of communication rounds on the CIFAR-10 dataset.

Then, we have

$$\mathbb{E}\left[\frac{1}{SK}\sum_{i\in\mathcal{S}_t}\left\|\boldsymbol{g}_i^{t,k}-\boldsymbol{g}^t\right\|\right] \leq 2\beta\left(\left(1+\frac{2}{\sqrt{K}}\right)\sigma+2\eta KL+\left(1+\frac{2N}{S}\right)\gamma L\right)$$
$$+2\beta\left(\frac{\sqrt{2}\sigma}{\sqrt{K}}+\sqrt{\frac{2S}{3N}}\eta KL\right)\left(1-\frac{S}{4N}\right)^{t-1}+\eta KL+2\gamma L.$$

Summing the above inequality over $t$, we have

$$\frac{1}{SKT}\sum_{t=1}^{T}\mathbb{E}\left[\sum_{i\in\mathcal{S}_t,k}\left\|\boldsymbol{g}_i^{t,k}-\boldsymbol{g}^t\right\|\right] \leq 2\beta\left(\left(1+\frac{2}{\sqrt{K}}\right)\sigma+2\eta KL+\left(1+\frac{2N}{S}\right)\gamma L\right)$$
$$+\frac{8N\beta}{ST}\left(\frac{\sqrt{2}\sigma}{\sqrt{K}}+\sqrt{\frac{2S}{3N}}\eta KL\right)+\eta KL+2\gamma L.$$

## C ADDITIONAL NUMERICAL RESULTS

**Experimental Settings:** We employ a convolutional neural network (CNN) with three convolutional layers and two fully connected layers for the EMNIST dataset, and a ResNet-18 architecture for CIFAR-10. The experimental framework involves 100 distributed clients with 10 clients participating randomly in each training round. We investigate both independent and identically distributed (i.i.d.) and non-i.i.d. data distributions. For i.i.d. scenarios, we implement uniform random data distribution across clients. To simulate realistic heterogeneity in non-i.i.d. settings, we apply a Dirichlet distribution Dir(1) for EMNIST, Dir(0.5) for CIFAR-10. The hyperparameters of all baselines, including learning rates, are optimized through comprehensive grid search.

### C.1 SIMULATIONS ON CIFAR-10 DATASET

Figure 3 presents the comparative analysis of test accuracy across different algorithms on the CIFAR-10 dataset, with subfigures 3a and 3b illustrating the performance under i.i.d. and non-i.i.d. data distributions, respectively. The observed patterns align with those demonstrated in Figure 1. Our proposed algorithms demonstrate superior performance compared to existing methods, including FedAvg, SCAFFOLD, and SCAFFOLD-M, both in terms of convergence rate and final test accuracy. Under non-i.i.d. conditions, while all algorithms exhibit increased performance volatility and reduced accuracy, the relative performance hierarchy remains consistent with the i.i.d. scenario, as shown in subfigure 3b.

Figure 4 illustrates the evolution of gradient norm $\|\nabla f(\boldsymbol{\theta}^t)\|$ for various algorithms on the CIFAR-10 dataset under both i.i.d. and non-i.i.d. data distributions. The results demonstrate that

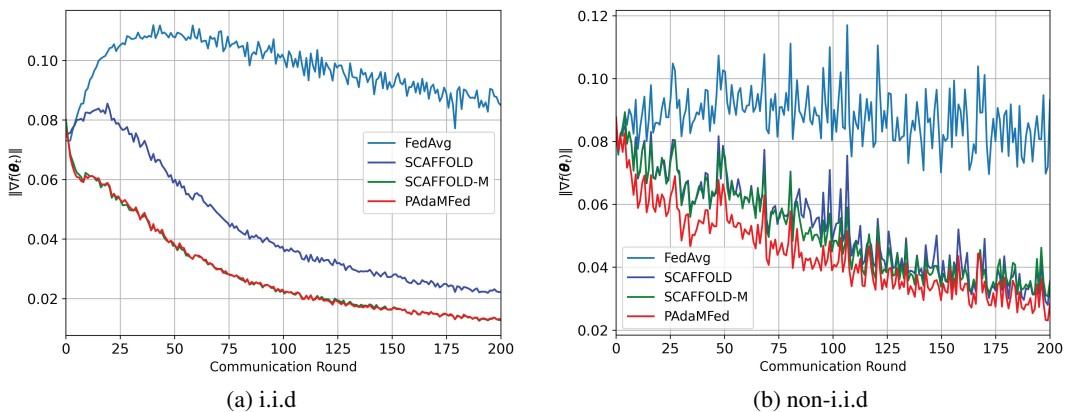

| (a) i.i.d | (b) non-i.i.d |

Figure 4: Gradient norm versus the number of communication rounds on the CIFAR-10 dataset.

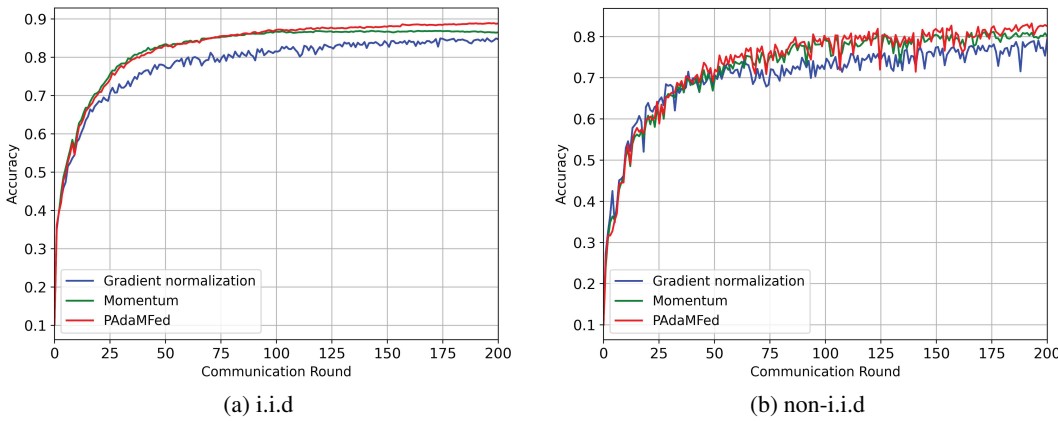

| (a) i.i.d | (b) non-i.i.d |

Figure 5: Ablation study versus the number of communication rounds on the CIFAR-10 dataset.

momentum-enhanced methods, specifically our proposed algorithm and SCAFFOLD-M, achieve more rapid gradient norm reduction compared to their non-momentum counterparts.

We further carry out the ablation study by isolating the effects of momentum and gradient normalization in Figure 5. The results demonstrate that incorporating SCAFFOLD with normalized gradient leads to degraded performance due to the loss of gradient magnitude information. Therefore, the momentum is essential in our algorithm design to maintain the descent direction by effectively aggregating gradients across clients and iterations.

