# OpenReview forum: "Problem-Parameter-Free Federated Learning"
_ICLR.cc/2025/Conference — ICLR 2025 Oral_

### Official Review · Reviewer_ZrV1 · 2024-10-29

**Soundness:** 3
**Presentation:** 3
**Contribution:** 3
**Rating:** 6
**Confidence:** 4

**Summary:**

This paper introduces PAdaMFed, a new federated learning (FL) algorithm that removes dependence on problem-specific parameters, a significant advancement for FL where hyperparameter tuning is often hindered by data heterogeneity and limited local dataset accessibility. By combining adaptive stepsizes and momentum, PAdaMFed aims to manage arbitrary data heterogeneity and partial client participation while achieving state-of-the-art convergence and communication complexities. Empirical results validate the theoretical benefits of PAdaMFed across multiple tasks, demonstrating robustness and scalability.

**Strengths:**

1. The parameter-free design in FL addresses a critical challenge, making this an interesting and important contribution.
2. Theoretical guarantees, including sample and communication complexities of \(O(\epsilon^{-4})\) and \(O(\epsilon^{-3})\) respectively for the standard version (with further improvements under variance reduction), are rigorously derived and well-supported.

**Weaknesses:**

1. The "problem-parameter free" claim is somewhat unclear. Although the authors suggest that PAdaMFed has no dependency on problem-specific parameters like the smoothness constant \(L\), it seems that implicit conditions on \(L\) might still affect the learning rate. For example, Assumptions A3 and A4 seem to require some condition on \(L\) related to \(K\) and \(T\), suggesting that the learning rate still indirectly depends on \(L\).

2. From my understanding, achieving a truly problem-parameter free algorithm would typically involve hyperparameter adaptivity based on problem characteristics. Simply incorporating variance reduction and momentum may not fully achieve this goal. In problem-parameter or hyper-parameter free algorithms in centralized learning, the key is to design the adaptivity of these hyper-parameters .

3. The algorithm requires communication of three vectors per round, which is substantial. Since communication often constitutes a major bottleneck in FL, this increase is likely not marginal, especially given the lack of evidence provided to support this claim.

4. I have concerns regarding the experiment results. The baselines, particularly FedAvg, appear to underperform, with an accuracy of only 0.8 on EMNIST IID data. The experimental section would be stronger if it included additional datasets and models, along with an ablation study to isolate the contributions of adaptive stepsizes and momentum, as proposed in the PAdaMFed, to the overall performance.

Overall, this paper targets a meaningful problem and makes a substantial contribution to algorithm design in federated learning, supported by theoretical rigor and empirical evidence. However, its novelty may be limited by reliance on existing algorithmic techniques and proof strategies (e.g., variance reduction and SCAFFOLD).

**Questions:**

1. I'd like to know the performance of these baselines, particularly FedAvg, which seems to underperform.
2. I suggest an ablation study to isolate the contributions of adaptive stepsizes and momentum in the PAdaMFed.

---

> ### Author Response · Authors · 2024-11-13
> **Point-to-Point Responses**
>
> **We sincerely appreciate the reviewer for recognizing our contributions and for the constructive comments. Our point-to-point responses to concerns on Weaknesses and Questions are given below.**
>
> **Reply to Weakness 1:**  We appreciate this thoughtful question about the "problem-parameter free" property. We would like to clarify the following two key points:
>
> 1) The term "problem-parameter free" refers to algorithms that do not require problem-specific parameters (such as smoothness constant $L$) for tuning in practice. In our algorithm, the learning rates are explicitly determined by only system-defined constants: the number of participating clients $S$, local update iterations $K$, and communication rounds $T$. These are configuration parameters that are known and set by the system administrator, rather than problem-specific parameters that would need to be estimated or tuned.
>
> 2) While our theoretical analysis includes assumptions involving the smoothness constant $L$ (e.g., Assumption 1), these assumptions are standard regularity conditions that establish the problem setting, rather than parameters needed for algorithm execution. Importantly, $L$ remains independent of both $K$ and $T$, and knowledge of $L$ is not required to run the algorithm. The practical implementation of PAdaMFed depends only on the system-defined constants ($S, K, T$), making it truly parameter-free from an operational perspective. Additionally, we would like to clarify that this paper only make three assumptions and  Assumptions 1 and 2 are related to $L$-smoothness. But the smoothness condition on $L$ is independent of both the number of clients $K$ and the communication rounds $T$. For reference, our Assumption 1 is cited below:
> >**Assumption 1.** *Given any $\xi$, the sample-wise loss function $F(\boldsymbol\theta; \xi)$ is $L$-smooth, i.e., $||\nabla F(\boldsymbol\theta; \xi)-\nabla F(\boldsymbol\delta; \xi)|| \leq L||\boldsymbol\theta-\boldsymbol\delta||$ for all $\boldsymbol\theta,\boldsymbol\delta \in \mathbb{R}^{d}$.*
>
> **Reply to Weakness 2:**  Thank you for the comment. Our algorithm achieves parameter-free operation through careful algorithmic design that incorporates automatic adaptivity. Specifically, the gradient normalization mechanism in Step 8 serves as the key adaptive component:
>
> 1) The normalization automatically adjusts step sizes based on the local optimization landscape, effectively providing adaptive learning rates without requiring manual tuning. This design ensures:
> - Larger steps in regions with small gradients where more aggressive exploration is beneficial;
> - Smaller steps in steep regions where careful progress is needed;
> - Uniform update magnitudes across clients, preventing any single client from having disproportionate influence.
>
> 2) This adaptive mechanism, combined with momentum and control variates, creates a truly self-tuning system where:
> - Update magnitudes are automatically calibrated based on local gradients;
> - The effective step size becomes proportional to the ratio between learning rate and gradient norm;
> - Client heterogeneity is naturally handled without requiring explicit bounds or parameters.
>
> Our approach thus achieves parameter-free operation through algorithmic adaptivity rather than manual tuning, making it robust across diverse federated learning scenarios.
>
> **Reply to Weakness 3:**  We appreciate the concern about communication efficiency. We would like to clarify the actual communication overhead of our algorithm:
>
> Uplink (client → server): Each client transmits two vectors per round: $\boldsymbol\theta_i^{t,K}$ and $\boldsymbol{c}_i^t$.
>
> Downlink (server → clients): Originally appears to require three vectors: $\boldsymbol\theta^t$, $\boldsymbol{c}^t$, and $\boldsymbol{g}^t$. However, we can optimize this by combining $\boldsymbol{c}^t$ and $\boldsymbol{g}^t$ into a single term $\beta\boldsymbol{c}^t + (1-\beta)\boldsymbol{g}^t$. This optimization reduces downlink communication to two vectors.
>
> With this implementation, our algorithm achieves the same communication overhead as SCAFFOLD, making it communication-efficient while maintaining its parameter-free advantages.

---

> ### Author Response · Authors · 2024-11-15
> **Point-to-Point Responses**
>
> **Reply to Weakness 4:** We appreciate these thoughtful suggestions about the experimental evaluation. Our responses to each point of the comment is listed below:
>
> 1) Regarding FedAvg's performance on EMNIST: Our results are consistent with previous studies in the literature. For instance,  references [1] and [2] reported similar convergence rates for FedAvg under comparable settings. In our simulation, the performance of all baseline algorithms were implemented under identical experimental conditions as our algorithms to ensure fair comparison.
> 2) We have enhanced our experimental results in the revised version (as provided in Appendix D.1) by:
> - Including additional experiments on CIFAR-10 dataset;
> - Conducting comprehensive ablation studies to analyze the individual contributions of adaptive stepsizes and momentum;
> - Providing detailed analysis of how each component affects the overall performance.
> 3) To ensure reproducibility and facilitate further research, we will release our code on GitHub upon acceptance.
>
> [1] J. Pei, W. Li and S. Mumtaz, "From Routine to Reflection: Pruning Neural Networks in Communication-efficient Federated Learning," in IEEE Transactions on Artificial Intelligence, 2024, doi: 10.1109/TAI.2024.3462300.
>
> [2] V. S. Mai, R. J. La, T. Zhang, Y. Huang and A. Battou, "Federated Learning With Server Learning for Non-IID Data," 2023 57th Annual Conference on Information Sciences and Systems (CISS), Baltimore, MD, USA, 2023, pp. 1-6.
>
> **Reply to the following comment in the Weaknesses:**
> >Overall, this paper targets a meaningful problem and makes a substantial contribution to algorithm design in federated learning, supported by theoretical rigor and empirical evidence. However, its novelty may be limited by reliance on existing algorithmic techniques and proof strategies (e.g., variance reduction and SCAFFOLD).
>
> **Reply:**  We sincerely appreciate the reviewer for recognizing our contributions. However, we would like to emphasize that our algorithm introduces significant innovations aimed at overcoming specific challenges of problem-specific parameter tuning in federated learning.
>
> 1) Our primary goal is to design a truly parameter-free federated learning algorithm that eliminates dependency on all problem-specific parameters while maintaining state-of-the-art convergence. This is achieved through the careful integration of three essential components:
> - Local gradient normalization for adaptive step-sizes;
> - Client-side momentum for heterogeneity bounding;
> - Control variates for "client-drift" control.
> 2) The theoretical analysis of our algorithm follows a significantly different path from SCAFFOLD's proof technique. Our analysis requires novel theoretical tools to handle the interplay between gradient normalization, momentum, and control variates, while establishing convergence guarantees without requiring knowledge of problem parameters.
> 3) Our algorithms also eliminate the need for data heterogeneity bounds, which is an important advancement than the SCAFFOLD algorithm.
> 4) The variance reduced version PAdaMFed-VR is an extension of our original algorithm PAdaMFed to further accelerate its convergence. Even without variance reduction, PAdaMFed demonstrates state-of-the-art performance compared to existing non-variance-reduced algorithms.
>
> This paper advance the field by providing a more robust and adaptable solution to federated learning challenges. We appreciate your recognition of the meaningful problem we address and the substantial contributions we make.
>
>
> **Reply to Question 1:** Thank you for this question. Our answer can refer to the Reply to Weakness 4.
>
> **Reply to Question 2:**  We appreciate the suggestion to conduct an ablation study. By isolating the effects of adaptive step sizes and momentum, we can better understand their individual contributions to PAdaMFed's performance. Figure 5 of the revised manuscript presents our comprehensive ablation analysis examining test accuracy and step size robustness with respect to these two key components (momentum and gradient normalization). The results demonstrate that momentum enhances convergence speed by effectively aggregating descent directions across clients and iterations. Meanwhile, gradient normalization facilitates adaptive step size adjustment, enabling parameter-free tuning of our algorithm.
>
> **Thank you once again for your thoughtful review and constructive feedback.**

---

> > ### Comment · Reviewer_ZrV1 · 2024-11-26
> >
> > Thank you for the rebuttal; it has addressed most of my questions. However, one point remains unclear: the major contribution claimed in the paper, the algorithm is the first to be problem-parameter-free in FL. Could you clarify whether the learning rate of the algorithm is entirely independent of the smoothness parameter L? Specifically, I’d like to understand how A3 and A4 hold in the proof (Appendix).

---

> > > ### Author Response · Authors · 2024-11-27
> > >
> > > Dear Reviewer ZrV1,
> > >
> > > **Thank you for your thoughtful feedback and for acknowledging that our rebuttal has addressed most of your questions.**
> > > Regarding your point about the learning rate's independence from the smoothness parameter $L$ and the clarity of how A3 and A4 hold in the proof, we would like to provide further clarification.
> > >
> > > **1. Stepsizes in Our Algorithm:**
> > >
> > > Our algorithm involves three stepsizes:
> > > - **Momentum coefficient:** $\beta = \sqrt{\frac{SK}{T}}$;
> > > - **Global learning rate:** $\gamma = \frac{(SK)^{1/4}}{T^{3/4}}$;
> > > - **Local learning rate:** $\eta = \frac{1}{K\sqrt{T}}$.
> > > These stepsizes are explicitly defined by system-defined constants $(S,K,T)$ and do not depend on the smoothness parameter $L$.
> > >
> > > **2. How A3 and A4 Are Derived:**
> > >
> > > A3 and A4 are derived directly by plugging these stepsize definitions ($\beta, \gamma, \eta$) into the bounds provided by Lemmas 3 and 4, respectively. Specifically:
> > >
> > > - In **Lemma 3**, the bound on $\frac{1}{T} \sum_{t=0}^{T-1} \mathbb{E} ||\nabla f(\theta^t) - g^t||$ depends on terms involving $L$, which arise naturally due to the smoothness assumption. However, the stepsizes ($\beta, \gamma, \eta$) themselves are **independent of $L$**, and the error terms involving $L$ are controlled by the adaptive decay of $\gamma =\frac{(SK)^{1/4}}{T^{3/4}}$.
> > > - Similarly, in **Lemma 4**, the bound on $\frac{1}{SKT} \sum_{t=0}^{T-1} \sum_{i \in \mathcal{S}_t, k} \mathbb{E} ||g_i^{t, k} - g^t||$ involves terms with $L$, but these arise from the analysis of the gradient variance and are not tied to the choice of $\beta, \gamma,$ or $\eta$. The convergence is ensured by appropriately scaling the stepsizes with respect to $T$, independent of $L$.
> > >
> > > As a result, no explicit knowledge of $L$ is required for the stepsizes, making the algorithm **problem-parameter-free** in this regard.
> > >
> > > **We hope this explanation resolves your concern, and we would be happy to provide further clarification if needed.**
> > >
> > > **Thank you again for your valuable feedback and we would be grateful if you could improve your score if all your concerns are addressed.**

---

> > > > ### Comment · Reviewer_ZrV1 · 2024-11-28
> > > >
> > > > Thank you for the response. My main concern remains the major contribution claimed in this paper: that it is the first problem-parameter-free approach in FL. In fact, there are many problem parameters beyond smoothness $L$, and it is well-known that normalization methods can naturally adapt to smoothness (e.g., https://parameterfree.com/2023/06/19/adapting-to-smoothness-with-normalized-gradients). When the proposed PAdaMFed algorithm is reduced to a centralized setting (e.g., one client with one local step), it essentially becomes normalized SGD with momentum. From my perspective, it would not be accurate to label this as problem-parameter-free. Please correct me if I was wrong.
> > > >
> > > > While I can appreciate the contributions in algorithm design and theoretical analysis, the claim of being the first problem-parameter-free algorithm in FL feels quite misleading, especially for those familiar with parameter-free optimization.

---

> ### Author Response · Authors · 2024-11-28
>
> **Thank you for your follow-up comment and for pointing out your concerns. We would like to clarify the following points to address the issues raised:**
>
> 1) **Difference from Normalized Gradient Methods:**
> The reference provided by the reviewer (https://parameterfree.com/2023/06/19/adapting-to-smoothness-with-normalized-gradients) focuses on the deterministic setting, where exact gradients are available at each step. In contrast, our work addresses the stochastic setting inherent to federated learning (FL), where only noisy, stochastic gradients are available at each update. This distinction is critical, as it has been formally shown (e.g., reference [1], Table 1) that **normalized stochastic gradient descent cannot converge to the optimal solution even in the centralized case**. Our contribution lies in overcoming this limitation by proposing a novel algorithm that achieves convergence in this more challenging stochastic environment.
>
> [1] Yang, Junchi, et al. "Two sides of one coin: the limits of untuned SGD and the power of adaptive methods." Advances in Neural Information Processing Systems 36 (2024).
>
> 2) **Challenges Unique to Federated Learning:**
> Federated learning introduces distinct challenges that are fundamentally different from the centralized case considered in normalized SGD methods. The presence of multiple local updates in FL exacerbates issues like **client drift**, especially under data heterogeneity. These challenges **cannot be addressed by a simple reduction to a one-step centralized case**. To mitigate client drift, we introduce a novel integration of control variates with momentum. This contribution is not only algorithmically innovative but also requires non-trivial theoretical analysis to account for the interactions between control variates and momentum in a federated setting. Our algorithm also exhibits another important innovation in the elimination of explicit heterogeneity bounds, which have been a standard requirement in prior federated learning literature.
> 3) **Definition of Problem-Parameter-Free:**
> We use the term "problem-parameter-free" in the commonly accepted sense, where it describes an algorithm that avoids explicit dependence on problem-specific parameters such as the smoothness constant or Lipschitz constant. Our method adheres to this definition, as it **does not require tuning or even knowledge of such parameters**, which is a significant departure from traditional FL algorithms. While normalized gradient methods can adapt to smoothness in the deterministic setting, their limitations in stochastic and federated settings render them insufficient for addressing the challenges we solve.
>
> In summary, while we appreciate the reviewer’s concerns and the reference provided, we believe our work makes significant contributions by addressing the unique challenges of the stochastic and federated setting. **Our approach is fundamentally different from normalized gradient descent methods in deterministic optimization**, as we overcome their known limitations in stochastic and federated scenarios. We would be happy to further clarify any remaining concerns and thank you again for your valuable feedback.

---

> > ### Comment · Reviewer_ZrV1 · 2024-12-02
> >
> > I appreciate the authors' responses. While the algorithm design and theoretical analysis presented in the paper are solid and compelling, I suggest refraining from claiming the algorithm as the 'first Problem-Parameter-Free FL algorithm,' as this term is somewhat ambiguous. I have raised my score accordingly.

---

> > > ### Author Response · Authors · 2024-12-02
> > >
> > > Dear Reviewer ZrV1,
> > >
> > >    Thank you very much for your reply and for recognizing our contributions.  We are grateful for your valuable suggestion and will revise the claim "first Problem-Parameter-Free FL algorithm" throughout our paper. Thank you again for raising your score and for your constructive comments.
> > >
> > >
> > > Sincerely,
> > >
> > > Authors of the paper

---

### Official Review · Reviewer_pGAa · 2024-10-29

**Soundness:** 3
**Presentation:** 3
**Contribution:** 2
**Rating:** 8
**Confidence:** 3

**Summary:**

The paper studies the convergence of federated averaging algorithms with momentum. Its main contribution is to establish the convergence and convergence rate of these algorithms when the step-sizes and momentum parameters in the local and aggregation algorithms are tuned independently of problem-specific parameters, such as the Lipschitz constant of the gradients. Instead, these parameters depend solely on the number of local and global iterations and the number of nodes sampled in each iteration.

**Strengths:**

The topic is important and well motivated. It is indeed a problem that for ensuring theoretical convergence one typically needs parameters like Lipschitz constant on the gradient, which are typically not known.

The paper is well written and the results look correct, at least the main steps in the proofs look correct.

**Weaknesses:**

I have some concerns with the setup. Many algorithms already permit diminishing step-sizes that do not require knowledge of the Lipschitz constant of the gradient, such as step-decay or similar approaches, which also come with theoretical guarantees. These methods have the added advantage that step-sizes can start relatively large and decrease over time, while in this approach, the step-size is inversely based on the total number of global and local iterations, which may be large. As a result, the step-sizes in this method are consistently small. There is no comparison to such methods here.

The experiments are not reproducible, the code is not provided and there is no information about hyper parameter tuning or other details for other algorithms, except that a grid search has been used.

**Questions:**

Since the number of sampled nodes is used in the parameter selection, what to do if the number of sampled nodes changes between iterations?

In the numerical results, why only considering accuracy? I get that accuracy, or some similar Machine Learning metrics are important to evaluation for Machine Learning applications. However, all the theory is related to optimization metrics, the size of the gradient norm. It would be good to also show how the numerical results align with the theoretical results in the paper.

In line 8 of Algorithm 1, why do you take scale the gradient direction with the gradient norm? This means that you don't use the magnitude of the gradient, only the direction of the gradient. This is probably why you don't need Lipschitz constant of the gradient, even in deterministic optimziation algorithms such algorithms converge without using Lipschitz constants, but if we don't use gradient magnitude then convergence rate will be worse worse and compatible to sub-gradient methods, since we are not exploiting the smoothness.

Given that for your algorithm you did not do any grid search to tune hyperparameters, while grid-search was used for the other algorithms, I am a bit surprised how much better results you get. In my experience, the hyperparameters obtain from the theory are usually not good in practice, usually one can find much better parameters from doing grid search. Would this be the case for your work?

In Table 1, it would be good if you could also include the complexity bounds.

---

> ### Author Response · Authors · 2024-11-14
> **Point-to-Point Responses**
>
> **We sincerely appreciate the reviewer for recognizing our contributions and for the constructive comments. Our point-to-point responses to concerns on Weaknesses and Questions are given below.**
>
> **Reply to Weakness 1:** Thank you for the comment.  To the best of our knowledge, our algorithm is the first to be truely problem-parameter-free in FL, regardless of whether diminishing or constant step sizes are employed. In our approach, the gradient normalization mechanism compensates for conservative step sizes, which provides two key benefits:
> - It automatically adjusts the effective step size based on the local optimization landscape;
> - It compensates for conservative nominal step sizes through adaptive scaling.
>
> The effectiveness of this approach is validated by both our theoretical analysis and empirical results, where our parameter-free algorithm achieves competitive or superior performance compared to manually-tuned baselines. This demonstrates that careful algorithmic design can overcome the potential limitations of conservative step sizes.
>
> **Reply to Weakness 2:**  Thank you for the comment.
> 1) We will release our complete codebase upon acceptance via GitHub,  including thorough documentation, running scripts, and configuration files.
> 2) All baselines are trained using the same model architecture as our algorithm: (1) a convolutional neural network (CNN) with three convolutional layers and two fully connected layers for image classification on the EMNIST dataset, and (2) a long short-term memory (LSTM) model for textual sentiment analysis on the IMDB dataset. The gradient search ranges from 1e-1 to 1e-5. Moreover, we have presented the performance of all baseline algorithms versus stepsize in Fig 2 for the image classification task, and the test accuracy in Fig 1 of all those baselines corresponds to the best stepsizes shown in Fig. 2.
> 3) We have also conducted additional experiments on the CIFAR-10 dataset using a ResNet-18 architecture, where similar results further corroborate our findings.
>
> **Reply to Question 1:**  Thank you for the insightful question regarding varying client participation. While our current theoretical analysis assumes a fixed number of participating clients $S$ per round for analytical tractability, our algorithm can be naturally adapted to accommodate variations in client participation. For scenarios with fluctuating client participation, we can define a variable $S_{min}$ to represent the minimum expected number of participants per round. The analysis remains valid by replacing $S$ with $S_{min}$.
>
>
> **Reply to Question 2:** Thank you for this valuable suggestion. Test accuracy indeed serves as the primary measure of generalization performance, which is a core objective in machine learning applications. While optimization metrics (e.g., gradient norm) can indicate convergence during training, they do not necessarily reflect model performance on unseen data. Nonetheless, we agree that optimization metrics provide useful insights into algorithmic behavior. In the revised version, we have included gradient norm trajectories in Fig. 4 of the revised version.
>
> **Reply to Question 3:**  Thank you for the comment.
>
> 1) The gradient normalization in Line 8 enables adaptive step-sizes of our algorithm by automatically adjusting to the local optimization landscape. While the normalization only focus on the gradient direction, our algorithm maintains competitive convergence through two complementary mechanisms:
> - The momentum term accumulates gradient information across iterations, helping to preserve magnitude-related information over time
> - The control variates help reduce variance and maintain alignment with the global objective
>
> 2) It's important to note that our convergence analysis relies on the Lipschitz constant of the gradient (which is acutally the smoothness parameter $L$ in our paper). The gradient normalization, combined with our carefully designed control variates and momentum, ensures that we fully exploit the problem structure without requiring explicit knowledge of these constants. Our theoretical guarantees show that the design of our algorithm still maintains start-of-the-art convergence properties while achieving problem-parameter free operation.

---

> > ### Author Response · Authors · 2024-11-14
> > **Point-to-Point Responses**
> >
> > **Reply to Question 4:**  Thank you for the comment.  The strong performance of our algorithm without grid search can be attributed to two key design elements that provide inherent adaptivity:
> >
> > 1) Gradient normalization: Gradient normalization serves as an adaptive learning rate scheme, automatically adjusting step sizes based on the local optimization landscape. This design automatically allows larger steps in regions with small gradients (where more aggressive exploration is beneficial) and smaller steps in steep regions (where careful progress is needed). This per-step adaptivity can potentially achieve better performance than fixed learning rates, even when the latter are optimized through grid search.
> >
> > 2) Momentum: Our momentum-based update design helps accelerate convergence while maintaining stability. The momentum term provides two key benefits: it helps overcome local irregularities in the loss landscape by accumulating gradients over multiple iterations, and it accelerates progress in directions of consistent gradient agreement.
> >
> > These adaptive mechanisms enable our algorithm to achieve strong performance without manual tuning, as they continuously optimize the learning dynamics throughout the training process.
> >
> > **Reply to Question 5:**   Thank you for this valuable suggestion.  We have included the complexity bounds in Table 1 in the revised manuscript.
> >
> > **Thank you once again for your thoughtful review and constructive feedback.**

---

> > ### Comment · Reviewer_pGAa · 2024-11-26
> >
> > Thanks for the response. I am mostly happy with response and I am okay by raising my score by 1. However, I have a problem with the statement "our algorithm is the first to be truely problem-parameter-free in FL", this is clearly wrong. If you do FedAve with one local step per iteration, then you get an FL algorithm that is equivalent to SGD, which is well known to converge for diminishing step-sizes like 1/t, not depending on problem parameters. Similar for other FL algorithms. If you want improved convergence rate then that is another story and I agree that the paper provide some value there.

---

> > > ### Author Response · Authors · 2024-11-26
> > >
> > > Dear Reviewer pGAa,
> > >
> > > Thank you for your thoughtful feedback and for considering raising your score. We appreciate your remarks and would like to clarify the statement regarding our algorithm being "the first to be truly problem-parameter-free in FL," as well as address the specific concerns raised.
> > >
> > > We acknowledge that algorithms like FedAvg with one local step per iteration, combined with a diminishing step-size schedule such as $\eta_t=\frac{1}{t}$ are indeed problem-parameter-free in a certain sense. However, we aim to highlight a critical limitation in such approaches when applied to **nonconvex problems**, which is the focus of our work. Specifically, for nonconvex optimization, diminishing step sizes like $\eta_t=\frac{\eta}{\sqrt{t+1}}$ (commonly used for ensuring convergence) can lead to **problematic exponential terms** in the convergence bounds, as shown in reference [1].
> > >
> > > **Key Points from Reference [1]:**
> > >
> > > 1) Reference [1] demonstrates that when using diminishing step sizes $\eta_t=\frac{\eta}{\sqrt{t+1}}$, if $\eta>\frac{1}{L}$ (where
> > > $L$ is the Lipschitz constant of the gradient), a disastrous exponential term of the form $(4e)^{\eta^2 L^2 - 1}$ appears in the convergence bound. This term grows exponentially with $\eta$, making the convergence extremely sensitive to the choice of $\eta$.
> > >
> > > 2) Furthermore, reference [1] proves in its **Theorem 2** that such an exponential term is **unavoidable**, even in the deterministic setting, when using diminishing step sizes.
> > >
> > > This result implies that while SGD (or FedAvg with one local step) can converge with diminishing step sizes, the convergence behavior heavily depends on the choice of $\eta$. Specifically, choosing a step size $\eta$ that is too large (even slightly exceeding $\frac{1}{L}$) can lead to divergence or extremely poor convergence rates. Tuning $\eta$ to avoid this exponential term requires explicit knowledge of $L$, thereby introducing a dependency on problem-specific parameters.
> > >
> > > **Contributions of Our Work:**
> > >
> > > In contrast, our algorithm avoids such sensitivity to problem-specific parameters like $L$, thanks to the gradient normalization mechanism. This mechanism ensures that the step sizes are adjusted adaptively at each iteration, eliminating the need for manually tuning $\eta$ or relying on explicit problem parameters. As a result, our algorithm achieves start-of-the-art convergence rates without requiring prior knowledge of problem-specific constants like $L$.
> > >
> > >
> > > **Clarification of Our Statement:**
> > >
> > > We acknowledge that our statement "our algorithm is the first to be truly problem-parameter-free in FL" may have been overly broad. We will revise this in the manuscript to clarify that our contribution lies in developing a **problem-parameter-free algorithm that achieves state-of-the-art convergence rates for nonconvex problems**, without suffering from the exponential term highlighted in reference [1]. This refinement will ensure the claim is precise and aligns with the specific context of our work.
> > >
> > > **Thank you again for your feedback, and we hope this clarification addresses your concerns.**
> > >
> > >
> > > [1]  Yang, Junchi, et al. "Two sides of one coin: the limits of untuned SGD and the power of adaptive methods." Advances in
> > >       Neural Information Processing Systems 36 (2024).

---

### Official Review · Reviewer_xnsX · 2024-11-01

**Soundness:** 3
**Presentation:** 2
**Contribution:** 3
**Rating:** 8
**Confidence:** 3

**Summary:**

This paper proposes an algorithm called PAdaMFed, where the hyperparameters don't depend on the problem-specific parameters such as smoothness constant, stochastic gradient variance bound, etc. PAdaMFed applies a client-drift control mechanism similar to SCAFFOLD but also uses normalized updates at the client level. It also has a momentum-based update rule at the global level. A variance-reduced version of PAdaMFed, called PAdaMFed-VR is also proposed. PAdaMFed and PAdaMFed-VR have communication complexities of $\mathcal{O}(\epsilon^{-3})$ and $\mathcal{O}(\epsilon^{-2})$, respectively, for converging to an $\epsilon$ stationary point. The convergence results don't rely on data heterogeneity bounds, which is also interesting. Empirical results on EMNIST and IMDB (in the appendix) show the efficacy of the proposed method.

**Strengths:**

**1.** This paper seems to be the first one to provide an algorithm whose hyperparameters are fully independent of the problem-specific parameters such as smoothness constant, stochastic gradient variance bound, etc. However, I'm not up to speed on all the relevant literature.

**2.** It is interesting that the results don't rely on any kind of heterogeneity bound.

**3.** Empirical results show that the proposed method is better than SCAFFOLD.

**Weaknesses:**

**1.** It is not clear to me what exactly is enabling the algorithm to work with hyperparameters that don't depend on problem-specific parameters. This aspect should be explained better. For instance, does normalization in step 8 of Algorithm 1 enable the hyperparameters to be independent of the smoothness constant?

**2.** How exactly the variation reduction step in PAdaMFed-VR leads to variance reduction should be explained more precisely. The current discussion in Section 3.2 is not very satisfactory. Is this step of PAdaMFed-VR inspired by the update rule of STORM [1]? Also since the relative weights of the gradients at $\theta_i^{t,k}$ and $\theta_i^{t-1}$ are different ($1$ and $1-\beta$, respectively), it'd be helpful to have some discussion on how $\beta$ should be chosen to reduce the variance and also retain a sufficient amount of client drift control (w/ weight $\beta$).

**3.** The empirical results show that PAdaMFed is better than SCAFFOLD even after the hyperparameters of SCAFFOLD are tuned. However, from what I understood, the derived convergence results don't indicate this. Can the authors please explain this?

---

[1]: Ashok Cutkosky and Francesco Orabona. "Momentum-based variance reduction in non-convex SGD". *Advances in neural information processing systems, 32, 2019*.

**Questions:**

**1.** Is there any intuition for why there is no requirement for any kind of heterogeneity bound for deriving the results of this paper?

---

> ### Author Response · Authors · 2024-11-14
> **Point-to-Point Responses**
>
> **We sincerely appreciate the reviewer for recognizing our contributions and for the constructive comments. Our point-to-point responses to concerns on Weaknesses and Questions are given below.**
>
> **Reply to Weakness 1:**  Thank you for this valuable suggestion. We have added more explanations in Section 4.2 of our paper on how our algorithm achieves hyperparameter independence. The reviewer is right that the gradient normalization in Step 8 is indeed the key mechanism enabling our algorithm to operate without problem-specific parameters. This gradient normalization serves as an adaptive learning rate scheme, automatically adjusting step sizes based on the local optimization geometry — taking larger steps when gradients are small and smaller steps when gradients are large. It also provides us the convenience on quantifying the distance between consecutive models in our theoretical analysis, maintaining that $||\theta_i^{t, k+1}-\theta_i^{t,k}||= \eta $ for all $i, k, t$.
>
> Momentum and control variables are also indispensable to achieve the satisfactory performance of our algorithm. Momentum helps accelerate convergence while maintaining stability. First, it helps overcome local irregularities in the loss landscape by accumulating gradients over clients and iterations; Second, it accelerates progress in directions of consistent gradient agreement. Control variates align local updates with the global objective, reducing variance in gradient estimates and ensuring more consistent updates across heterogeneous client data. Collectively, these techniques ensure that the stepsizes are independent of problem-specific parameters such as smoothness parameters and heterogeneity bounds, simplifying the tuning process and enhancing robustness across diverse FL applications.
>
> **Reply to Weakness 2:**  Thank you for the comment.  The variance reduction technique used by PAdaMFed-VR is derived from the update rule of STORM. Specifically, the term $g_i^{t,k}$ in PAdaMFed-VR can be expressed as: $$g_i^{t,k} = \beta\left( \nabla F(\theta_i^{t,k};\xi_i^{t,k}) - \boldsymbol{c}_i^{t-1} + \boldsymbol{c}^{t-1} \right) + (1-\beta)  \boldsymbol{g}^{t-1} + (1-\beta) \left( \nabla F(\theta_i^{t,k};\xi_i^{t,k}) - \nabla F(\theta^{t-1};\xi_i^{t,k})\right).$$
> The first two terms are utilized in the non-variance-reduced PAdaMFed, while the effect of variance reduction is due to the last difference term, similar to STORM.
> We thank the reviewer for pointing out this vagueness in our paper. We have added a clearer explanation in the revised version.
>
>
> **Reply to Weakness 3:** Thank you for the comment. The strong performance of our algorithm can be attributed to two key design elements that provide inherent adaptivity:
>
> 1) Gradient normalization: Gradient normalization serves as an adaptive learning rate scheme, automatically adjusting step sizes based on the local optimization landscape. This design automatically allows larger steps in regions with small gradients (where more aggressive exploration is beneficial) and smaller steps in steep regions (where careful progress is needed). This per-step adaptivity can potentially achieve better performance than fixed learning rates, even when the latter are optimized through grid search.
>
> 2) Momentum: Our momentum-based update design helps accelerate convergence while maintaining stability. The momentum term provides two key benefits: it helps overcome local irregularities in the loss landscape by accumulating gradients over multiple iterations, and it accelerates progress in directions of consistent gradient agreement.
>
> These adaptive mechanisms enable our algorithm to achieve strong performance without manual tuning, as they continuously optimize the learning dynamics throughout the training process.
>
>
>
> **Reply to Question 1:** Thank you for the question. The use of momentum helps eliminate the requirement for data heterogeneity bounds. At each global round, the momentum is updated as follows:
> $$g^{t} = \frac{\beta}{SK}\sum_{i\in S_t}(\sum_{k=1}^K\nabla F(\theta_i^{t,k};\xi_i^{t,k})-\mathbf{c}_i^{t-1})+\beta\mathbf{c}^{t-1}+(1-\beta)g^{t-1}$$
>
> $$~~~=\frac{1}{SK}\sum_{i\in S_t}\sum_{k=1}^Kg_i^{t,k}.$$
> This accumulates the gradient directions across clients and iterations, providing an "anchoring" effect that effectively mitigates the “client-drift” phenomenon. In the extreme case where $\beta=0$, all clients remain synchronized in their local updates, eliminating the drift caused by data heterogeneity in the vanilla FEDAVG. By appropriately choosing the coefficient $\beta$, our algorithm achieves state-of-the-art convergence while removing the necessity for data heterogeneity assumptions.
>
> **Thank you once again for your thoughtful review and constructive feedback.**

---

> > ### Comment · Reviewer_xnsX · 2024-11-25
> > **Response**
> >
> > Thanks for the rebuttal. I'm mostly satisfied with the responses. So I will increase my score.

---

> > > ### Author Response · Authors · 2024-11-25
> > >
> > > Dear Reviewer xnsX,
> > >
> > >    Thank you very much for recognizing our contributions and for the reply.
> > >
> > >
> > > Sincerely,
> > > Authors of the paper

---

### Official Review · Reviewer_XEQX · 2024-11-01

**Soundness:** 2
**Presentation:** 3
**Contribution:** 3
**Rating:** 8
**Confidence:** 3

**Summary:**

The paper introduces PAdaMFed, a federated learning algorithm that uses momentum and adaptive learning rates to address client heterogeneity, similar to SCAFFOLD-M. The method is problem-parameter-free and does not require parameter tuning. The paper provides a convergence bound, without the standard assumption about gradient dissimilarity.

**Strengths:**

1- The paper studies important problems. Both data heterogeneity and hyperparameter tuning are existing issues for FL systems. The method has problem-independent hyperparameters which makes it more useful in practice.

2- There are not many works in the literature that study problem-independent parameters; this work is one of the first.

3- The convergence bound and analysis is novel, and has minimal assumptions. The authors proved convergence without assumptions about gradient dissimilarity, a standard assumption that most of the literature's works require.

**Weaknesses:**

1- The convergence bound is not optimal compared to SCAFFOLD-M. Also, for the case when  $K$ (the number of local updates) is a function of $T$ (total rounds), a fair comparison would involve the equivalent formulation for SCAFFOLD-M under the same conditions.

2- Empirical results are limited. Experiments for different architectures and datasets can help improve the paper and show its effectiveness.

3- Missing related work: There are some similarities between your work and [1] which is not cited. Can you compare your method and results with this work.

Minor: Most of related works uses $\|\nabla f(x)\|^2 \le \epsilon$ and not  $\|\nabla f(x)\| \le \epsilon$ for the definition of $\epsilon$-stationary and this formulation makes the paper a bit confusing.

[1] Li, Jiaxiang, et al. "Problem-Parameter-Free Decentralized Nonconvex Stochastic Optimization." arXiv preprint arXiv:2402.08821 (2024).

**Questions:**

1- Can you recover the optimal bound with the extra assumption and different learning rates?

2- Can you design an experiment in which the gradient dissimilarity assumption does not hold and other methods fail to converge?

suggestion: since $c^{t-1}$ and $g^{t-1}$ are used only as $\beta c^{t-1} + (1- \beta) g^{t-1}$ you can only send the weighted sum and not each one to the clients and have the same communication per round as SCAFFOLD.

---

> ### Author Response · Authors · 2024-11-20
> **Point-to-Point Responses**
>
> **We sincerely appreciate the reviewer for recognizing our contributions and for the constructive comments. Our point-to-point responses to concerns on Weaknesses and Questions are given below.**
>
> **Reply to Weakness 1:** We sincerely appreciate the reviewer for pointing this out. SCAFFOLD-M achieves a convergence rate of $\frac{1}{T}\sum_{r=0}^{T-1}\mathbb{E}[||\nabla f(\theta^t)||^2] \leq \mathcal{O}\left(\sqrt{\frac{L\Delta\sigma^2}{SKT}} + \frac{L\Delta}{T}\left(1 + \frac{N^{2/3}}{S}\right)\right)$. While this bound is theoretically superior to our method by a factor of $\frac{N^{2/3}}{S}$ in the second term, the overall convergence behavior is primarily governed by the first term. Therefore, our comparative analysis focuses on the dominant first term, which represents the primary bottleneck in practical convergence.
>
> In the revised version, we have explicitly incorporated this theoretical distinction to provide a more comprehensive comparison.
>
> **Reply to Weakness 2:**  We sincerely appreciate the reviewer for bringing [1] to our attention. We have added a comprehensive comparison of our work with [1] in the revised version.  While both papers address problem-parameter free optimization, our work additionally need to handle the unique challenges inherent in federated learning. The key distinction lies in addressing client drift—a critical issue that emerges from multiple local updates and data heterogeneity in federated settings. Compared to [1], our main technical contributions are:
> - The novel integration of control variates with momentum to mitigate client drift, which requires sophisticated theoretical analysis due to their non-trivial interactions.
> - The elimination of explicit heterogeneity bounds that were previously required in federated learning literature.
>
> **Reply to the Minor Weakness:** We sincerely appreciate the reviewer for pointing this out. We have clarified this choice throughout the manuscript. Our analysis uses $||\nabla f(\theta^t)||$ instead of $||\nabla f(\theta^t)||^2$ as this emerges naturally from our theoretical framework for establishing problem-parameter-free convergence. While conventional federated learning analyses typically employ the squared gradient norm, our novel proof technique specifically requires direct analysis of the gradient norm to achieve parameter-free convergence guarantees.
>
> To facilitate comparison with existing literature, we have demonstrated how our results translate to the conventional $||\nabla f(\theta^t)||^2$ metric in the revised version through the following relationship:
> $$ \frac{1}{T} \sum_{t=1}^{T-1} \mathbb{E} \|\nabla f(\theta^t) \| = \frac{1}{T} \sum_{t=0}^{T-1} \mathbb{E} \sqrt{\|\nabla f(\theta^t) \|^2} \leq \frac{1}{T} \sum_{t=0}^{T-1} \sqrt{\mathbb{E} \|\nabla f(\theta^t)\|^2} \leq \sqrt{\frac{1}{T} \sum_{t=0}^{T-1} \mathbb{E} \|\nabla f(\theta^t)\|^2 } $$
> where the first and second inequalities utilizes Jensen’s inequality as the square root function is concave. Therefore, our results can be translated to the conventional $||\nabla f(\theta^t)||^2$ metric by squaring both sides of our convergence bounds.
>
>
> **Reply to Question 1:**  Thank you for your question. In our current analysis, we have achieved state-of-the-art convergence rates by relying on assumptions of smoothness (Assumptions 1 and 2) and stochastic gradient variance (Assumption 3). While these assumptions are sufficient for our theoretical guarantees, incorporating additional assumptions could potentially enhance the convergence bounds. If an extra assumption, such as strong convexity, were introduced, it's likely that we could derive tighter convergence bounds. These modifications might allow us to recover or even improve upon the optimal convergence rates observed in other algorithms. However, our focus has been on maintaining a robust and broadly applicable framework with minimal assumptions. We are open to exploring these possibilities in future work to further enhance the theoretical and practical performance of our algorithm.

---

> > ### Author Response · Authors · 2024-11-20
> > **Point-to-Point Responses**
> >
> > **Reply to Question 2:** Thank you for raising this important point. The gradient dissimilarity assumption critically impacts both stepsize selection and convergence guarantees. When gradient dissimilarity bounds are incorrectly estimated, algorithms may employ learning rates that fall outside their theoretically guaranteed performance regions, potentially leading to degraded convergence behavior.
> >
> > This phenomenon is empirically demonstrated in Figure 2, which compares test accuracy across different stepsize choices. The results clearly show that our algorithm maintains stable performance across a substantially wider range of stepsizes compared to baseline methods. This robustness stems from eliminating the dependency on gradient dissimilarity bounds, thus avoiding the practical challenges of stepsize tuning under heterogeneous data distributions.
> >
> > **Reply to Suggestion:** Thank you for this valuable suggestion. We have revised our algorithm to transmit the combined message $\beta\boldsymbol{c}^t + (1-\beta)\boldsymbol{g}^t$ instead of sending $\boldsymbol{c}^t$ and $\boldsymbol{g}^t$ separately.
> >
> > **Thank you once again for your thoughtful review and constructive feedback.**

---

> > > ### Comment · Reviewer_XEQX · 2024-11-25
> > >
> > > Thanks for your responses.
> > >
> > > Some additional notes:
> > >
> > > 1) > Therefore, our results can be translated to the conventional $||\nabla f(\theta^t)||^2$  metric by squaring both sides of our convergence bounds.
> > >
> > > This is the other way around. The conventional bounds can translated to your setting, and not the other way around.
> > >
> > > 2) > This robustness stems from eliminating the dependency on gradient dissimilarity bounds.
> > >
> > > The results in Figure 2 holds also for the i.i.d case, So I don't think the difference is due to gradient dissimilarity.
> > >
> > >
> > > Other than these points, you have addressed my main concerns, so I raise my score.

---

> > > > ### Author Response · Authors · 2024-11-25
> > > >
> > > > Dear reviewer XEQX,
> > > >
> > > >     Thank you very much for your reply and the additional notes. Your comments has greatly improved our paper, especially the additional note 1.
> > > >
> > > > Sincerely,
> > > >
> > > > Authors of the paper

---

### Official Review · Reviewer_A59W · 2024-11-11

**Soundness:** 3
**Presentation:** 3
**Contribution:** 3
**Rating:** 8
**Confidence:** 4

**Summary:**

This paper proposed a new federated learning algorithm called PAdaMFed that is problem-parameter free. The main idea is to combine SCAFFLOD and momentum. The authors also designed a modified version of PAdaMFed to reduce the variance, called PAdaMFed-VR  The authors showed that PAdaMFed-VR can achieve state-of-the-art convergence performance. The performance of PAdaMFed and PAdaMFed-VR is also verified using experiments.

**Strengths:**

1. The proposed PAdaMFed is independent of the problem parameters such as gradient divergence.

2. The author also designed PAdaMFed-VR to reduce the variance of the federated learning algorithm.

3. The authors showed the convergence upper bound of PAdaMFed and PAdaMFed-VR analytically.

**Weaknesses:**

1. The idea of PAdaMFed is a direct extension of SCAFFOLD with momentum considered at each client.

2. The challenge in the proof is unclear.

**Questions:**

1. What are the challenges in the proof compare to the proof in the SCAFFOLD algorithm?

2. The assumptions used in this paper are similar to those used in proving the convergence rate of the SCAFFOLD algorithm, so the gain in terms of convergence analysis must coming from using momentum and variance reduction. Please provide a detailed explanation on the gain of the proposed algorithms. Please provide a detailed comparison in terms of the convergence rate/communication complexity between PAdaMFed, PAdaMFed-VR, the SCAFFOLD algorithm and related algorithms as the Table 2 in the SCAFFOLD paper.

Karimireddy, Sai Praneeth, Satyen Kale, Mehryar Mohri, Sashank Reddi, Sebastian Stich, and Ananda Theertha Suresh. "Scaffold: Stochastic controlled averaging for federated learning." In International conference on machine learning, pp. 5132-5143. PMLR, 2020.

**Details Of Ethics Concerns:**

N/A.

---

> ### Author Response · Authors · 2024-11-15
> **Point-to-Point Responses**
>
> **We sincerely appreciate the reviewer for recognizing our contributions and for the constructive comments. Our point-to-point responses to concerns on Weaknesses and Questions are given below.**
>
> **Reply to Weakness 1:**  Thank you for the comment. While we appreciate the observation regarding SCAFFOLD, we respectfully disagree that PAdaMFed is merely a direct extension with momentum. Our work has a fundamentally different objective and makes distinct technical contributions:
>
> 1) Our primary goal is to design a truly parameter-free federated learning algorithm that eliminates dependency on all problem-specific parameters while maintaining state-of-the-art convergence. This is achieved through the careful integration of three essential components:
> - Local gradient normalization for adaptive step-sizes;
> - Client-side momentum for heterogeneity bounding;
> - Control variates for "client-drift" control.
>
> 2) The theoretical analysis of our algorithm follows a significantly different path from SCAFFOLD's proof technique. Our analysis requires novel theoretical tools to handle the interplay between gradient normalization, momentum, and control variates, while establishing convergence guarantees without requiring knowledge of problem parameters.
>
> While both algorithms use control variates, PAdaMFed's parameter-free nature and incorporation of adaptive mechanisms represent fundamental innovations rather than incremental extensions. Notably, our algorithms also eliminate the need for data heterogeneity bounds, which is an important advancement than the SCAFFOLD algorithm.
>
> **Reply to Weakness 2 and Question 1:** Thank you for the comment. The proof of our algorithm presents several unique challenges compared to SCAFFOLD's analysis:
> 1) Handling normalized gradients: Our use of gradient normalization introduces new mathematical complexities. We need to carefully track how normalization affects the relationship between consecutive model updates. Special techniques are required to analyze the interaction between normalized gradients and momentum.
> 2) Momentum analysis: The introduction of momentum terms creates additional coupling between iterations. We need to bound the accumulated effects of momentum across multiple updates. The interaction between momentum and control variates requires novel analysis techniques.
> 3) Parameter-free guarantees: Proving convergence without relying on problem-specific parameters requires different analytical tools. We need to show that gradient normalization effectively compensates for the lack of parameter-dependent step sizes. The analysis must demonstrate that convergence holds across different problem settings without parameter tuning.
>
> These challenges required developing new proof techniques that differ significantly from SCAFFOLD's analysis approach.
>
> **Reply to Question 2:**  Thank you for the comment.
> 1) We have enhanced our explanation on the gain of our proposed algorithms in the revised manuscript. Specifically, the gain of our PAdaMFed algorithm stems from the synergistic integration of three indispensable components:
> - Gradient normalization serves as an adaptive learning rate scheme, automatically adjusting step sizes based on the local optimization landscape. This design automatically allows larger steps in regions with small gradients (where more aggressive exploration is beneficial) and smaller steps in steep regions (where careful progress is needed).
> - Client-side momentum helps accelerate convergence while maintaining stability. First, it helps overcome local irregularities in the loss landscape by accumulating gradients over clients and iterations; Second, it accelerates progress in directions of consistent gradient agreement.
> - Furthermore, control variates align local updates with the global objective, reducing variance in gradient estimates and ensuring more consistent updates across heterogeneous client data.
>
> Collectively, these techniques ensure that the stepsizes are independent of problem-specific parameters such as smoothness parameters and heterogeneity bounds, simplifying the tuning process and enhancing robustness across diverse FL applications.
>
>
> 2) We have provided a detailed comparison in terms of the convergence rate/communication complexity between PAdaMFed, PAdaMFed-VR, the SCAFFOLD algorithm and related algorithms as the Table 2 in the SCAFFOLD paper.
>
> **Thank you once again for your thoughtful review and constructive feedback.**

---

> > ### Comment · Reviewer_A59W · 2024-11-27
> >
> > I appreciated the authors' detailed response of my questions. I raised my score to 8.

---

> > > ### Author Response · Authors · 2024-11-28
> > >
> > > Dear Reviewer A59W,
> > >
> > > Thank you very much for recognizing our contributions and for the reply.
> > >
> > >
> > > Sincerely,
> > >
> > > Authors of the paper

---

### Author Response · Authors · 2024-11-25
**Follow-Up: Request for ICLR Rebuttal Response**

Dear Reviewers,

Thank you again for taking the time to review our paper and provide your valuable feedback. As the rebuttal period comes to a close, we wanted to kindly check if our clarifications in the rebuttal have satisfactorily addressed the concerns raised in your initial review. Your feedback is invaluable to us.

If so, we respectfully request you to consider updating your review score based on our responses. However, if you have any remaining questions or need further clarification from us, please do not hesitate to let us know.

Once again, we sincerely appreciate your time and consideration. Your prompt response would be greatly appreciated.

Sincerely,

Authors of the paper

---

### Meta-Review · Area_Chair_RxdN · 2024-12-19

**Metareview:**

This paper proposes a novel federated learning algorithm, PAdaMFed, designed to be "problem-parameter-free". Specifically, as the authors clarify, the stepsize and momentum parameters required for the method depend only on system configurations, such as the total number of iterations and the number of clients, rather than on problem-specific parameters like smoothness.

The *parameter-free* property is achieved through normalized gradient updates performed locally, which ensure that the stepsize and momentum parameters do not depend on problem-specific parameters like smoothness. Additionally, the algorithm incorporates control variate-based drift correction (as in SCAFFOLD) and momentum-based variance reduction (similar to STORM), enhancing its robustness and efficiency in the federated learning setting.

The authors provide a theoretical analysis, proving convergence to stationary points for smooth, potentially non-convex functions.

Overall, this paper addresses an important and timely problem in federated learning and substantially contributes to algorithm design. The proposed method is not only theoretically sound but also practically relevant, as it eliminates the need for problem-specific parameter tuning. This work will likely inspire further research on adaptive and robust algorithms for federated learning.

---
Comment:
I do not agree with the authors' response to reviewer XEQX (resp. their added footnote 1 on page 6). The provided inequality appears to go in the wrong direction. A transfer of the results might be possible when considering $\min_{t} ||\nabla f(\theta^t)||^2$, but the result does not necessarily hold for the average gradient norm squared.

**Additional Comments On Reviewer Discussion:**

The discussion with the reviewers clarified the term "parameter-free" and addressed the claim of being the "first parameter-free" algorithm, which the authors acknowledged might not be accurate and promised to correct in the revised version. Additionally, the reviewers suggested including additional references that could be relevant for the final version.

---

### Decision · Program_Chairs · 2025-01-22

Accept (Oral)